# Autophagy mitigates ethanol-induced mitochondrial dysfunction and oxidative stress in esophageal keratinocytes

Prasanna M. Chandramouleeswaran[1☯], Manti Guha[2☯], Masataka Shimonosono[2☯], Kelly A. Whelan[3], Hisatsugu Maekawa[2], Uma M. Sachdeva[2,4], Gordon Ruthel[5], Sarmistha Mukherjee[6,7], Noah Engel[1], Michael V. Gonzalez[8], James Garifallou[8], Shinya Ohashi[9], Andres J. Klein-Szanto[10], Clementina A. Mesaros[11], Ian A. Blair[11], Renata Pellegrino da Silva[8], Hakon Hakonarson[8], Eishi Noguchi[12], Joseph A. Baur[6,7], Hiroshi Nakagawa[2]*

1 Division of Gastroenterology, Department of Medicine, Perelman School of Medicine, University of Pennsylvania, Philadelphia, Pennsylvania, United States of America, 2 Herbert Irving Comprehensive Cancer Center, Columbia University, New York, New York, United States of America, 3 Fels Institute for Cancer Research and Molecular Biology, Lewis Katz School of Medicine, Temple University, Philadelphia, Pennsylvania, United States of America, 4 Division of Thoracic Surgery, Massachusetts General Hospital, Boston, Massachusetts, United States of America, 5 Department of Biomedical Sciences, Mari Lowe Center for Comparative Oncology, School of Veterinary Medicine, University of Pennsylvania, Philadelphia, Pennsylvania, United States of America, 6 Institute for Diabetes, Obesity and Metabolism, Perelman School of Medicine, University of Pennsylvania, Philadelphia, Pennsylvania, United States of America, 7 Department of Physiology, Perelman School of Medicine, University of Pennsylvania, Philadelphia, Pennsylvania, United States of America, 8 Center for Applied Genomics, Children's Hospital of Philadelphia, Philadelphia, Pennsylvania, United States of America, 9 Department of Therapeutic Oncology, Graduate School of Medicine, Kyoto University, Shogoin, Sakyo-ku, Kyoto, Japan, 10 Histopathology Facility, Fox Chase Cancer Center, Philadelphia, Pennsylvania, United States of America, 11 Translational Biomarkers Core, Center of Excellence in Environmental Toxicology, University of Pennsylvania, Philadelphia, Pennsylvania, United States of America, 12 Department of Biochemistry and Molecular Biology, College of Medicine, Drexel University, Philadelphia, Pennsylvania, United States of America

☯ These authors contributed equally to this work.
* hn2360@cumc.columbia.edu

**Data Availability Statement:** RNA-seq data area available via GEO (GSE143479). The other relevant

## Abstract

During alcohol consumption, the esophageal mucosa is directly exposed to high concentrations of ethanol (EtOH). We therefore investigated the response of normal human esophageal epithelial cell lines EPC1, EPC2 and EPC3 to acute EtOH exposure. While these cells were able to tolerate 2% EtOH for 8 h in both three-dimensional organoids and monolayer culture conditions, RNA sequencing suggested that EtOH induced mitochondrial dysfunction. With EtOH treatment, EPC1 and EPC2 cells also demonstrated decreased mitochondrial ATPB protein expression by immunofluorescence and swollen mitochondria lacking intact cristae by transmission electron microscopy. Mitochondrial membrane potential (ΔΨm) was decreased in a subset of EPC1 and EPC2 cells stained with ΔΨm–sensitive dye MitoTracker Deep Red. In EPC2, EtOH decreased ATP level while impairing mitochondrial respiration and electron transportation chain functions, as determined by ATP fluorometric assay, respirometry, and liquid chromatography-mass spectrometry. Additionally, EPC2 cells demonstrated enhanced oxidative stress by flow cytometry for mitochondrial superoxide (MitoSOX), which was antagonized by the mitochondria-specific antioxidant MitoCP.

data are within the manuscript and its Supporting Information files.

**Funding:** This study was supported by the following NIH (https://www.nih.gov/) Grants: K01DK103953 and R01DK121159 (KAW), P01CA098101 (HN), U54CA163004 (HN), R01DK114436 (HN), and R01AA026297 (PMC, MS, MG, and HN), P30DK050306, P30ES013508 University of Pennsylvania Center of Excellence in Environmental Toxicology (HN, CAM, IB), P30CA013696. EN is a recipient of the W. W. Smith Charitable Trust (http://www.wwsmithcharitabletrust.org/)grant (#C0608). No funders play any role in the study design, data collection and analysis, decision to publish, or preparation of the manuscript.

**Competing interests:** The authors have declared that no competing interests exist.

Concurrently, EPC1 and EPC2 cells underwent autophagy following EtOH exposure, as evidenced by flow cytometry for Cyto-ID, which detects autophagic vesicles, and immunoblots demonstrating induction of the lipidated and cleaved form of LC3B and downregulation of SQSTM1/p62. In EPC1 and EPC2, pharmacological inhibition of autophagy flux by chloroquine increased mitochondrial oxidative stress while decreasing cell viability. In EPC2, autophagy induction was coupled with phosphorylation of AMP activated protein kinase (AMPK), a cellular energy sensor responding to low ATP levels, and dephosphorylation of downstream substrates of mechanistic Target of Rapamycin Complex (mTORC)-1 signaling. Pharmacological AMPK activation by AICAR decreased EtOH-induced reduction of ΔΨm and ATP in EPC2. Taken together, acute EtOH exposure leads to mitochondrial dysfunction and oxidative stress in esophageal keratinocytes, where the AMPK-mTORC1 axis may serve as a regulatory mechanism to activate autophagy to provide cytoprotection against EtOH-induced cell injury.

## Introduction

Alcohol is a major lifestyle-related risk factor for many human diseases including oral and esophageal cancers. The oral cavity and the esophagus are unique sites that are directly exposed to the high concentrations of ethanol (EtOH) present in alcoholic beverages. Alcohol consumption results in cellular DNA damage in humans [1, 2], which is in part attributable to acetaldehyde, the primary product of EtOH breakdown (oxidation) that induces DNA adduct formation in the esophagus [3, 4]. Additionally, alcohol-induced DNA damage may be mediated by reactive oxygen species (ROS). As the major site of acetaldehyde breakdown, mitochondria may produce excessive ROS when cellular acetaldehyde overwhelms the mitochondrial capacity for acetaldehyde clearance [5]; however, the extent to which alcohol-induced cell injury involves mitochondria in human esophageal keratinocytes remains unknown.

One mechanism by which cells may cope with alcohol-induced cell injury and energy crisis is via autophagy. As an evolutionarily conserved homeostatic cellular process, autophagy provides stress-induced cytoprotection by promoting degradation and clearance of cytoplasmic substrates including damaged or dysfunctional mitochondria. Autophagy inhibition exacerbates alcohol-induced acute and chronic liver dysfunction *in vivo* [6, 7]. Adenosine monophosphate (AMP)-activated protein kinase (AMPK) is the master sensor of cellular energy status. It is activated by the serine/threonine kinase LKB1 upon decreased intracellular ATP levels, which is one indicator of mitochondrial dysfunction [8]. AMPK activates autophagy via inhibition of mTORC1 to permit activation of pro-autophagic proteins such as ULK1 [9, 10]. Mitochondrial dysfunction and oxidative stress activate AMPK to promote autophagy-mediated cell survival under several established conditions including nutrient deprivation [11, 12], however, little is known as to how esophageal keratinocytes may survive alcohol-induced toxicity.

Here, we investigated alcohol-induced cellular and molecular changes in esophageal keratinocytes. We find that the AMPK-mTORC1 axis may be involved in autophagy activation that permits cell survival under conditions of EtOH-induced mitochondrial dysfunction.

## Materials and methods

### Reagents and biologicals

All chemicals were purchased from Sigma-Aldrich, MO, USA unless otherwise noted.

### Animal experiments

C57/BL6 mice (8–12 weeks old male and female) (Jackson laboratory, ME, USA) received humane care and underwent procedures according to a protocol approved by Institutional Animal Care and Use Committees (IACUC) at the University of Pennsylvania and Columbia University. In acute alcohol challenge experiments, mice were subjected to oral gavage with a single bolus of 5 g/kg of 31.5% ethanol (Decon Laboratories, PA, USA). Control mice received an equal volume of phosphate-buffered saline (PBS, Gibco, MD, USA). Mice were sacrificed 6 hours later and the esophageal epithelial cells were harvested as described previously [13].

### Cell culture and esophageal three-dimensional (3D) organoids

Immortalized normal human esophageal keratinocyte cell lines (EPC series: EPC1-hTERT, EPC2-hTERT and EPC3-hTERT, hereafter EPC1, EPC2 and EPC3, respectively) were cultured in fully supplemented Keratinocyte Serum-free media (KSFM, Thermo Fisher Scientific, MA, USA) as described [13, 14]. Cells were counted by Countess™ Automated Cell Counter (Thermo Fisher Scientific) where dead cells were stained with 0.2% Trypan Blue dye (Thermo Fisher Scientific). All cell lines were routinely validated negative for mycoplasma by MycoA-lert™ Mycoplasma Detection Kit (Lonza, Basel, Switzerland). EPC1 and EPC2 have been characterized extensively [15, 16] and authenticated by genetic profiling using polymorphic short tandem repeat loci (ATCC, Manassas, VA, USA). EPC3 was newly established as described previously [13, 14] from a de-identified healthy 35 year-old Japanese male who underwent routine screening endoscopy for research biopsies following informed consent under an Institutional Review Board protocol approved at Kyoto University (SO).

Esophageal 3D organoids were generated and characterized as described [15, 17]. Briefly, live cells were suspended in Matrigel basement membrane matrix (BD Biosciences, CA, USA) and seeded at 2000 cells per 50 μl Matrigel in each well of 24-well plates (Thermo Fisher Scientific) and grown in KSFM medium supplemented with 0.6 mM CaCl$_2$ (KSFMC) (Sigma-Aldrich). Organoid growth was monitored using phase-contrast images captured by the EVOS FL Cell Imaging System (Thermo Fisher scientific) or bright-field images captured by KEY-ENCE Fluorescence Microscope BZ-X800 (Keyence, Osaka, Osaka, Japan). Number of viable cells in monolayer culture was evaluated by WST-1 assay (Promega, Madison, WI, USA), according to the manufacturer's instructions. CellTiter-Glo® Luminescent Cell Viability assay (Promega) was used to evaluate number of viable cells in 3D organoids.

For EtOH treatment in monolayer culture, cells were seeded in 6-well plates or 100-mm dishes (for immunoblot analysis), and sub-confluent (~80%) cells were exposed to 0.01–80% (v/v) EtOH. 3D organoids were exposed to 0.2–2% EtOH in 24-well plates. 0.01–4% EtOH was prepared by serially diluting 100% EtOH in KSFM. 5–80% EtOH was prepared by diluting 100% EtOH in Dulbecco's phosphate-buffered saline (PBS). Control cells received KSFM only, except in the experiments where cells were briefly exposed for 15 seconds to 5–80% EtOH in PBS or PBS only (control) and thereafter grown in KSFM for up to 24 h. Empty wells were filled with EtOH and the plates were tightly sealed with PARAFILM® M (Sigma-Aldrich) to maintain alcohol saturation.

Chloroquine diphosphate (CQ, Sigma-Aldrich) and Mitochondria targeted carboxy-proxyl (Mito-CP, a gift of Dr. Balaraman Kalyanaraman, Department of Biophysics and Free Radical

Research Center, Medical College of Wisconsin, WI, USA) were reconstituted in water. 5-Aminoimidazole-4-carboxamide ribonucleotide (AICAR), an AMPK activator, was reconstituted in water at 1 mM.

## RNA-seq library preparation and sequencing

RNA was isolated as described previously [18]. To generate total RNA library with rRNA depletion, the TruSeq Stranded Total RNA library kit (Illumina, CA, USA) was utilized. Libraries were produced using liquid handler automation with the Sciclone NGSx Workstation (PerkinElmer, MA, USA). This procedure was started with rRNA depletion step with target-specific oligonucleotides with specialized rRNA removal beads which remove both cytoplasmic and mitochondrial rRNA from the total RNA. Following this purification, RNA was fragmented using a brief, high-temperature incubation. The fragmented RNA was then reverse transcribed into first-strand cDNA using reverse transcriptase and random primers. Second strand cDNA was generated using DNA polymerase which was then used in a standard TruSeq Illumina-adapter based library preparation. Library preparation consisted of four main steps: unique adapter-indexes were ligated to the RNA fragments, AmpureXP bead purification occurred to remove small library fragments, the libraries were enriched and amplified using PCR, and the libraries underwent a final purification using AmpureXP beads. Upon completion, library quality was assessed using an automated electrophoresis instrument, the PerkinElmer Labchip GX Touch, and qPCR using the Kapa Library Quantification Kit and Viia7 real-time PCR instrument. Libraries were diluted to the appropriate sequencer loading concentration and pooled accordingly to allow for the desired sequencing depth. RNA libraries were sequenced in one lane of the Illumina HiSeq2500 sequencer using the High Output v4 chemistry and paired-end sequencing (2x100bp).

## RNA-seq data analysis

RNA-seq reads were demultiplexed using the DRAGEN genome pipeline [19]. FASTQ files were aligned to hg19 reference using the STAR (v.2.6.1) aligner with default settings [20]. Generated BAM files were read into the R statistical computing environment. Gene counts were obtained using the Genomic Alignments package. Differential expression analysis was performed using the R/Bioconductor package DESeq2 which uses a negative binomial model. Analysis was performed using standard parameters with the independent filtering function enabled to filter genes with low mean normalized counts. The Benjamini-Hochberg adjustment was used to estimate the false discovery rate (Padj) and correct for multiple testing. FDR (Padj <0.01) was used to identify differentially expressed genes in each condition. These genes were then analyzed using the QIAGEN Ingenuity Pathways Analysis (IPA) software (QIAGEN, Redwood City, CA, USA) for functional annotation and to identify upregulated or downregulated pathways in each dataset [21].

## Immunofluorescence

Cells were grown on glass coverslips, fixed in 4% formaldehyde for 20 minutes, permeabilized with 0.1% Triton X-100 in Dulbecco's PBS and blocked with 5% bovine serum albumin for 1 hour. Cells were incubated with mouse anti-ATPB (3D5) (1:1000; ab14730; abcam USA) overnight at 4°C, and then with mouse-Cy2-conjugated secondary antibodies (1:100; Jackson ImmunoResearch, PA, USA) for 1 hour at room temperature. Nuclei were counterstained by 4',6-diamidino-2-phenylindole (DAPI, 1:10000; Thermo Fisher Scientific). Stained objects were imaged with a Leica SP5 FLIM confocal microscope.

### Transmission Electron Microscopy (TEM)

TEM was performed as described previously [18]. In brief, cells were fixed with 2.5% glutaraldehyde and 2% paraformaldehyde in 0.1 M cacodylate buffer (pH 7.4) at 4°C overnight. Samples were further processed at the Electron Microscopy Resource Laboratory at the University of Pennsylvania. Images were obtained using a JEOL 1010 electron microscope (JEOL USA, MA, USA) fitted with a Hamamatsu digital camera (Hamamatsu, Boulder CO, USA) and AMT advanced imaging software (Advanced Microscopy Techniques, MA, USA). Micrographs were assessed by two investigators (KAW and AJP) blinded to experimental conditions.

### Measurement of cellular ATP levels

Total cellular ATP levels were determined using the ATP Fluorometric Assay kit (Sigma) following manufacturer's recommendations. Briefly, $10^6$ cells were lysed using the ATP Assay Buffer, which releases cellular ATP by altering membrane permeability. Cell lysates were deproteinized using a10kD MWCO spin filter. To correct for background in samples, we used a sample blank by omitting the ATP Converter. The sample blank readings were subtracted from the sample readings. ATP-dependent fluorescence was measured on a Glomax Microplate Reader (Promega). ATP was calculated using appropriate ATP standard curve. All samples and standards were assayed in duplicate. Cells were pretreated with the pharmacologic compounds (as indicated in the figure): Chloroquine (CQ, autophagy inhibitor) for 32 h (24 h pretreatment), AICAR (AMPK activator) for 9 h (1 hour pretreatment).

### Liquid Chromatography-High Resolution Mass Spectrometry (LC-HRMS)

To determine tricarboxylic acid (TCA) cycle metabolites, 5 x $10^5$ EPC2-hTERT cells were seeded in 10 cm tissue culture dish and cultured in compete KSFM media. Sub-confluent cells were further cultured in regular KSFM media or media containing 2% EtOH for 8 h. Cells were washed twice in 0.9% NaCl and then 1000 μl ice cold methanol/water (4/1 v/v) was added together with a mixture of internal standards as previously described [22]. Cells were scraped and sonicated for 30 sec followed by centrifugation at 16,000 g for 10 min. Supernatants were moved to clean tubes and dried under nitrogen for quantification of polar metabolites after resuspension in 100 μl 5% sulphosalicylic acid. LC-HRMS was performed as previously described [23].

### High resolution respirometry

Cells were treated with either 0% or 2% EtOH for 8 h and were re-suspended at a concentration of 1.0 x $10^6$/ml in pre-warmed MIRO5 respiration buffer, containing 110 mM mannitol, 0.5 mM EGTA, 3 mM $MgCl_2$, 20 mM taurine, 10 mM $KH_2PO_4$, 60 mM K-lactobionate, 0.3 mM dithiothreitol, and 0.1% fatty acid-free bovine serum albumin at pH7.2. A standard substrate/inhibitor titration protocol was used for functional analysis of esophageal cell respiratory function [24]. The cells were subjected to the respirometer and after stabilization the baseline oxygen consumption were measured using high-resolution respirometry at 37°C with constant stirring on the Oxygraph-2k FluoRespirometer (Oroboros Instruments, Innsbruck, Austria), then were permeabilized with 3–4 μl of 1 μg/μl digitonin and allowed to stabilize for 5 min. Following stabilization, real-time oxygen concentration and flux data were continuously collected using substrates and inhibitors for the electron transport chain (ETC). Complex I-dependent respiration was induced by adding 10 mM pyruvate, 10 mM malate, and 1 mM adenosine diphosphate to the respiration chamber. In order to determine Complex II-

dependent respiration, 0.5 μM rotenone, a selective inhibitor of Complex I, was added followed by 10 mM succinate. 5 μM antimycin A was then added to inhibit complex III followed by 0.5 mM N,N,N',N'-Tetramethyl-p-Phenylenediamine and 2 mM ascorbate as artificial substrates for complex IV. This protocol was completed within 40 min. The data were analyzed using DatLab software 4.3 (Oroboros Instruments).

## Flow cytometry

Flow cytometry was performed using LSR II (BD Biosciences) and FlowJo software (Tree Star) as described previously [5, 25]. In brief, autophagosome (i.e. autophagic vesicles, AV) content was determined with Cyto-ID® fluorescent dye (ENZ-51031; Enzo Life Sciences, NY, USA). ROS were determined with MitoSOX™ for mitochondrial superoxide (M36008; 5 μM; Thermo Fisher Scientific). Cells were incubated with MitoTracker™ Green (1:20,000; M7514; Thermo Fisher Scientific) and MitoTracker™ Deep Red (1:50,000; M22426; Thermo Fisher Scientific) in KSFM at 37˚C for 30 min to determine mitochondrial mass and mitochondrial membrane potential, respectively. Geometric mean fluorescence intensity for MitoTracker™ Green and MitoTracker™ Deep Red were determined in the live cell fraction (DAPI-) for each specimen following subtraction of background fluorescence. Apoptosis was determined using the Annexin-V-FLUOS kit (11858777001; Sigma-Aldrich) according the manufacturer's instructions. Cell viability was determined using DAPI.

## Immunoblot analysis

Immunoblotting was performed as described previously [18]. The following primary antibodies were purchased from Cell Signaling Technology (Danvers, MA, USA) and used at the indicated dilutions: rabbit anti-LC3B (#2775) (1:1000), mouse anti-p62/SQSTM1 (D-3; sc-28359) (1:1000), rabbit anti-phospho-S6 (Ser$^{235/236}$; #2211 and #4858) (1:1000 for #2211 or 1:2000 for # 4858), mouse anti-S6 (#2317) (1:1000), rabbit anti-S6 (#2217) (1:1000), rabbit anti-phospho 4EBP1 (Thr$^{37/46}$; #9459 (1:1000), rabbit anti-4EBP1 (#9452) (1:1000), rabbit anti-phospho-AMPK (#2535) (1:1000), rabbit anti-AMPK (#2603) (1:1000) and mouse anti-β-Actin (AC-74; A5316; Sigma-Aldrich) (1:10000). Densitometry of resulting signals was performed with Image J (National Institutes of Health).

## RNA interference and transfection

SiRNA directed against ADH1B (s1061 and s1062) and *CYP2E1* (s3837 and s3838) or a non-target control sequence (SCR Silencer Select Negative Control #1) were purchased from Thermo Fisher Scientific. Reverse transfection was performed using Lipofectamine RNAi Max reagent (Thermo Fisher Scientific) as described previously [18].

## RNA isolation, cDNA synthesis, quantitative reverse-transcription polymerase chain reactions (qRT-PCR)

RNA isolation, cDNA synthesis and qRT-PCR were done using StepOnePlus™ Real-Time PCR System (Applied Biosystems) by TaqMan® Gene Expression Assays (Applied Biosystems) for *ADH1B* (s02511271_s1) and *CYP2E1* (Hs00559367_m1), and SYBR® Green PCR for human *ACTB* (5'-TTGCCGACAGGATGCAGAA-3' and 5'-GCCGATCCACACGGAGTACT-3'). Relative target mRNA level was normalized to β-actin (ACTB) as described [18]. All PCR reactions were performed in triplicate. The relative level of each mRNA was normalized to *ACTB* (β-actin) as internal controls.

## Statistical analyses

Data were analyzed as indicated using Prism 7.0 software (GraphPad, CA, USA). $p < 0.05$ was considered significant. Equal variance across groups being compared was confirmed by $F$-test. Data are representative of at least two independent experiments with similar results.

## Results

### Normal esophageal keratinocytes remain viable in the continuous presence of 2% EtOH in 3D organoid and monolayer culture conditions

Unlike internal organs such as the liver, during alcohol consumption, the esophageal epithelium is directly exposed to 2–40% (v/v) EtOH, much higher than the life-threatening blood alcohol level (~0.5% or 85 mM). A prior radiographic study in human subjects demonstrated that ingested radiolabeled substances can remain in the esophagus for up to 2 h following single bolus swallowing [26]. Admixed with saliva and mucus produced by esophageal glands, a relatively high concentration of EtOH may stay on the esophageal mucosal surface during and after alcohol consumption; however, the alcohol tolerance capacity of esophageal keratinocytes remains undefined.

We first determined the EtOH tolerance of the normal human esophageal epithelial cell lines EPC1, EPC2, and EPC3 in 3D organoid culture, a single cell-derived platform that recapitulates the normal proliferation-differentiation gradient of the esophageal epithelium [15, 16]. 3D organoids grew in the presence of 0.6% (100 mM) EtOH, albeit slower than untreated control organoids (Fig 1A, S1A Fig). When established 3D organoid structures were exposed to 0.2–2% EtOH, 2% EtOH suppressed cell viability by 40–50% at 24 h (Fig 1B and 1C) and decreased organoid size at 24 h (Fig 1D, S1B Fig). However, EtOH sensitivity appeared to be variable amongst esophageal epithelial cell lines. In established 3D organoids, 0.6% EtOH decreased cell viability of EPC3, but not EPC1 or EPC2, within 24 h (Fig 1B) while 2% EtOH decreased cell viability of EPC3 faster than EPC1 or EPC2 (Fig 1C). Of note, 2% PBS did not affect cell viability of all three esophageal cell lines tested (Fig 1B), suggesting that the potential effect of medium dilution by EtOH is negligible, if any. EPC3 cells overall displayed slower growth, with smaller organoid size than EPC1 or EPC2 even in the absence of EtOH (Fig 1A and 1D, S1 Fig).

In monolayer culture, EtOH, but not PBS (4%), had a similar inhibitory effect upon EPC1, EPC2 and EPC3 cell viability in a dose-dependent manner. However, 2% EtOH suppressed cell viability to a greater extent at 24 h (Fig 2A) than in 3D organoid culture conditions (Fig 1B and 1C). EPC3 cells were more sensitive to 1–2% EtOH than EPC1 or EPC2 at 24 h (Fig 2A), in agreement with the higher EtOH sensitivity of EPC3 cells seen in 3D organoids. This result was further corroborated by flow cytometry for DAPI-stained cells, where DAPI-positive dead cells were higher in EPC3 than in EPC1 or EPC2 following 2% EtOH exposure for 8 h (Fig 2B). Of note, 2% EtOH exposure alone did not increase the DAPI-positive cell rate at 8 h in EPC1 and EPC2 (Fig 2B). However, we detected early apoptotic cell death by flow cytometry for Annexin V staining in EPC1 and EPC2 cells continuously exposed to 3% EtOH for 8 h (S2 Fig). Additionally, we assessed approximately 10–15% of cell death was occurring (i.e. DAPI-positive cells in Fig 2B and PI-positive cells in S2A Fg) independent of EtOH exposure through experimental procedures (e.g. trypsinization, flow cytometry in single cell suspensions). When EPC2 cells were exposed to higher concentrations of EtOH relevant to most alcoholic beverages, >80–90% of cells tolerated up to 20% EtOH when exposure time was limited to 15 sec (S3 Fig), a standard time for a swallowed liquid to pass through the human esophageal lumen [27–29]. Based on the above observations, we concluded that 2% may be the highest

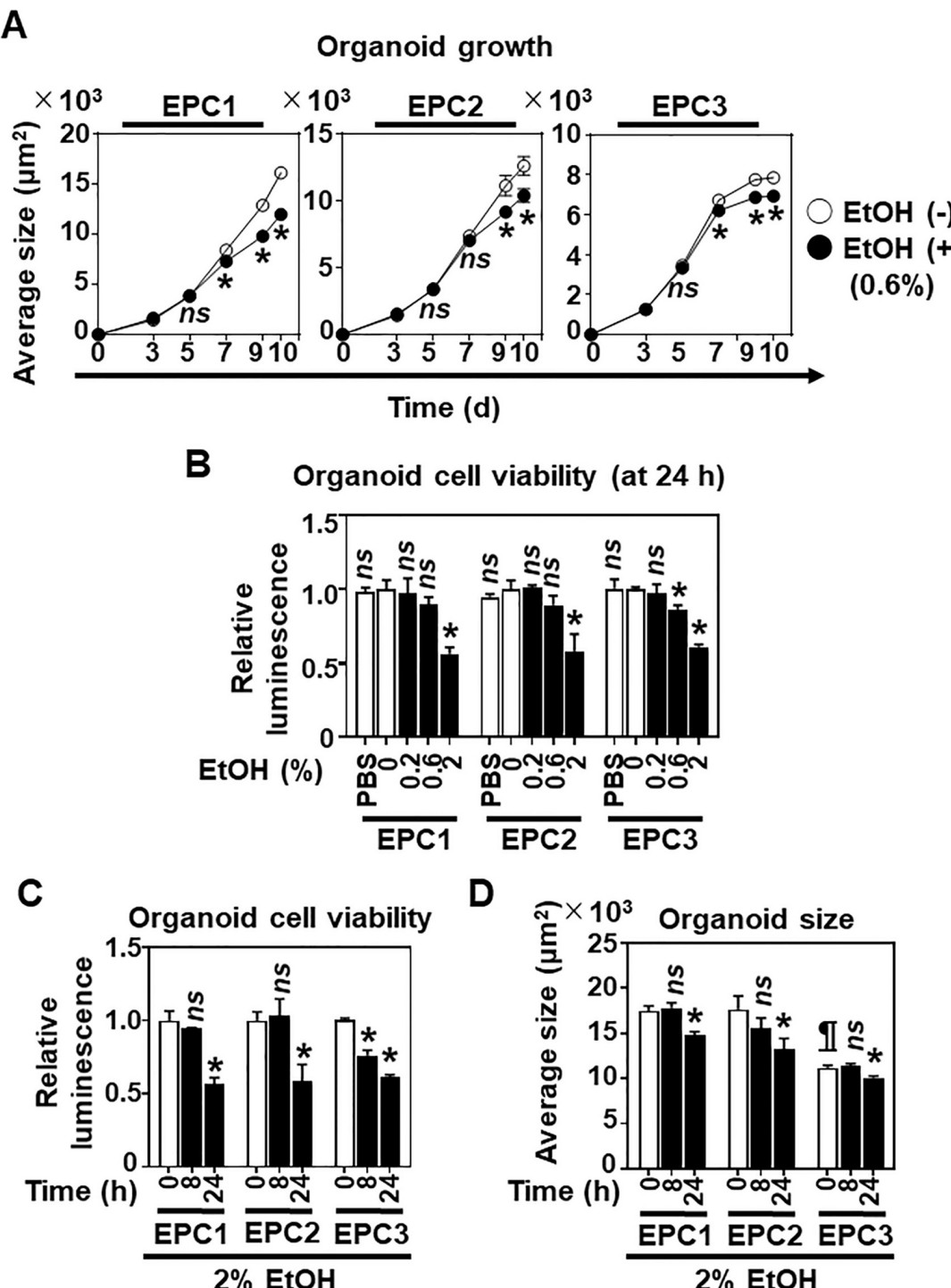

**Fig 1. EtOH inhibits cell growth and viability in esophageal 3D organoids.** Esophageal 3D organoids were generated from EPC1, EPC2 and EPC3 cells. **A**. 3D organoids were grown in the presence or absence of EtOH exposure (0.6%, 7 days from day 3 through day 10) in triplicate. Growth curve of 3D organoids was created by plotting average organoid sizes at the indicated time points. The size of individual organoids was determined as the area of each spherical structure in bright-field images (S1 Fig) captured by an automated microscope. At least 100 organoids were imaged per well at each time point. **B-D**. 3D organoids were first grown in the absence of EtOH for 9 days and then treated with or without EtOH. In **B**, organoids were treated in triplicate for 24 h with PBS (2%) or EtOH at indicated concentrations. PBS was used to assess the potential influence of medium dilution by EtOH. In **C** and **D**, 3D organoids were treated in triplicate with or without 2% EtOH for indicated time periods. In **B** and **C**, cell viability in 3D organoids was determined by CellTiter-Glo® assays. In **D**, average organoid size was determined as in

**A**. Data represent mean ± sem. n = 3 per condition. *, p<0.05 vs. EtOH (-); *ns*, not significant vs. EtOH (-) at indicated time points in **A**. *, p<0.05 vs. PBS or 0% EtOH; *ns*, not significant vs. PBS or 0% EtOH in **B**. *, p<0.05 vs. 0 h; *ns*, not significant vs. 0 h in **C**. *, p<0.05 vs. 0 h; *ns*, not significant vs. 0 h; and ¶, p<0.05 vs. EPC1 or EPC2 at 0 h in **D**. Student's t-test was used in **A**-**D**.

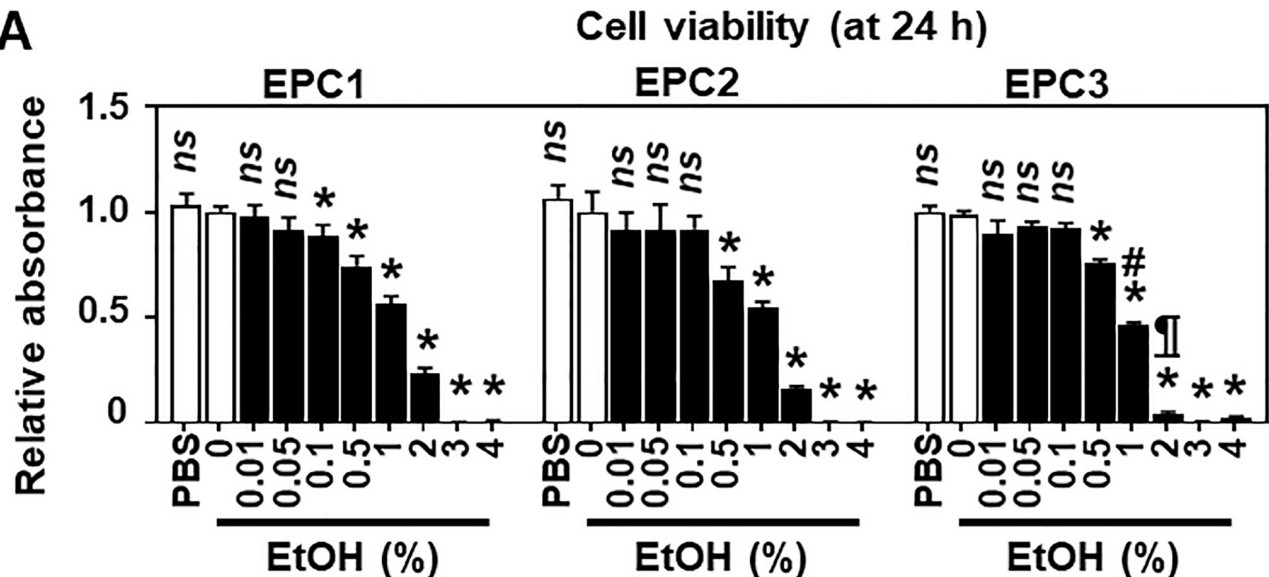

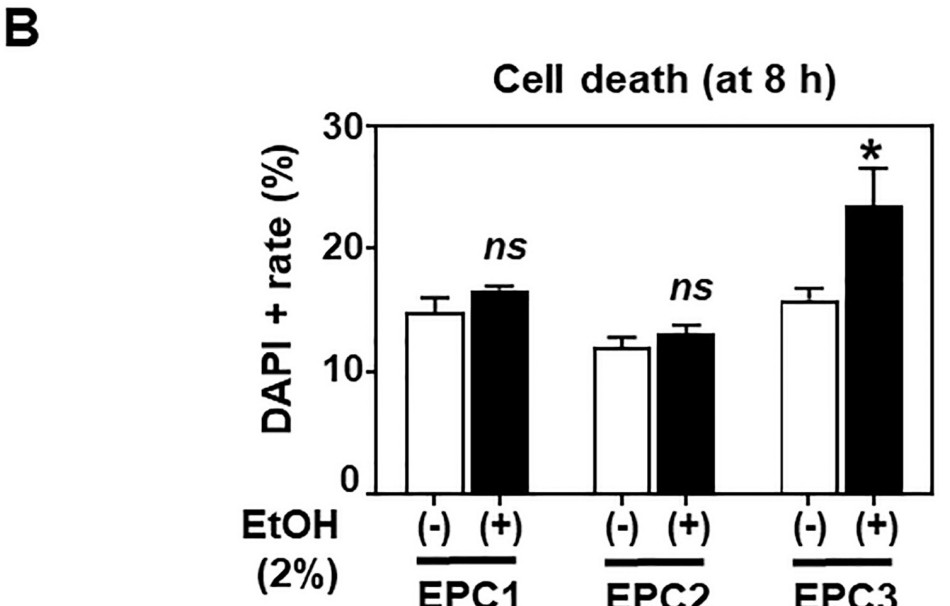

**Fig 2. EtOH inhibits esophageal cell viability in monolayer culture.** EPC1, EPC2 and EPC3 cells were grown to subconfluence and were treated with or without EtOH in monolayer culture to determine cell viability (**A**) and death (**B**). **A**. Cells were treated for 24 h with PBS (4%) or EtOH at indicated concentrations and subjected to WST-1 assays. PBS was used to assess the potential influence of medium dilution by EtOH. **B**. Cells were treated for 8 h with or without 2% EtOH and stained with DAPI for flow cytometry. Data present mean ± sem. n = 4 per condition in **A** and n = 3 per condition in **B**. *, p<0.05 vs. PBS or 0% EtOH; *ns*, not significant vs. PBS or 0% EtOH; #, p<0.05 vs. EPC1 or EPC2 treated with 1% EtOH; ¶, p<0.05 vs. EPC1 or EPC2 treated with 2% EtOH in **A**. *, p<0.05 vs. 0% EtOH; *ns*, not significant vs. EtOH (-) in **B**. Student's t-test was used in both **A** and **B**.

concentration of EtOH that the majority of normal esophageal keratinocytes may tolerate for at least 8 h when present continuously.

## RNA sequencing reveals mitochondria as a target for EtOH in esophageal keratinocytes

To gain molecular insights into early cellular changes associated with alcohol exposure in an unbiased manner, we performed RNA sequencing (RNA-seq) analysis of monolayer EPC1, EPC2 and EPC3 cells treated with or without 0.1% or 2% EtOH for 8 h. We chose an 8 h time point because the majority of cells (>70%) were deemed viable in monolayer culture (Fig 2B). RNA-seq yielded 32 million input reads per sample of which 24 million were uniquely mapped on average (S1 Table). 0.1% EtOH had minimal impact upon gene expression (q<0.01) as visualized by principal component analysis (PCA) (S4 Fig); however, 2% EtOH induced significant changes in 6985 genes with 3370 upregulated and 3615 downregulated (GSE143479). Canonical pathway analysis suggested downregulation of the TCA cycle as the foremost change induced by EtOH exposure (Fig 3A), indicating that EtOH exposure may lead to decreased mitochondrial metabolism. In agreement, gene set enrichment analysis (GSEA) indicated that those associated with mitochondrial structural and functional components including oxidative phosphorylation may be amongst the most negatively enriched sets of genes (Fig 3B and S2 Table). Additionally, EtOH downregulated significantly genes essential in cell-cycle regulation and DNA damage response pathways (ATM signaling and nucleotide excision repair) (Fig 3A). To investigate whether mitochondrial dysfunction may be induced by EtOH, we focused on EPC1 and EPC2 cells in further experiments because they were more similar in gene expression profile (S4 Fig) and EtOH sensitivity (Figs 1 and 2) than EPC3.

## EtOH affects mitochondrial structural integrity in esophageal keratinocytes

We evaluated how EtOH may influence mitochondrial structure and function in esophageal keratinocytes. Structural alterations in mitochondria are associated with alterations in mitochondrial function. First, we treated EPC1 and EPC2 with or without EtOH and performed immunofluorescence for the mitochondria-localized protein ATPB. High-resolution confocal imaging revealed a well-organized filamentous network of mitochondria in EtOH-untreated (control) cells (Fig 4A). EtOH exposure (2%, 8h) resulted in a decrease of the filamentous mitochondrial network and development of ring- or donut-shaped structures (Fig 4A). Such structures have been linked to mitochondrial energy crisis observed under hypoxic conditions [30]. We next analyzed mitochondrial ultrastructure by transmission electron microscopy (TEM). Intact mitochondria with prominent cristae were readily detectable in untreated (control) cells (Fig 4B). In EtOH-treated cells, we observed numerous swollen mitochondria lacking intact cristae (Fig 4B) that may be compatible with damaged mitochondrial structures. Of note, EtOH-treated cells displayed an increase in intracellular vacuolar structures (Fig 4B). Most of them were suspected to be involutional mitochondria, and some had a double membrane with electron dense contents, reminiscent of autophagic vesicles (AV) (Fig 4C).

## EtOH may affect mitochondrial membrane potential, mitochondrial respiration and metabolism to promote oxidative stress in esophageal keratinocytes

Stable mitochondrial membrane potential ($\Delta\Psi$m) is a salient feature of healthy mitochondria with intact electron transport chain (ETC) functions. $\Delta\Psi$m reduction or mitochondrial

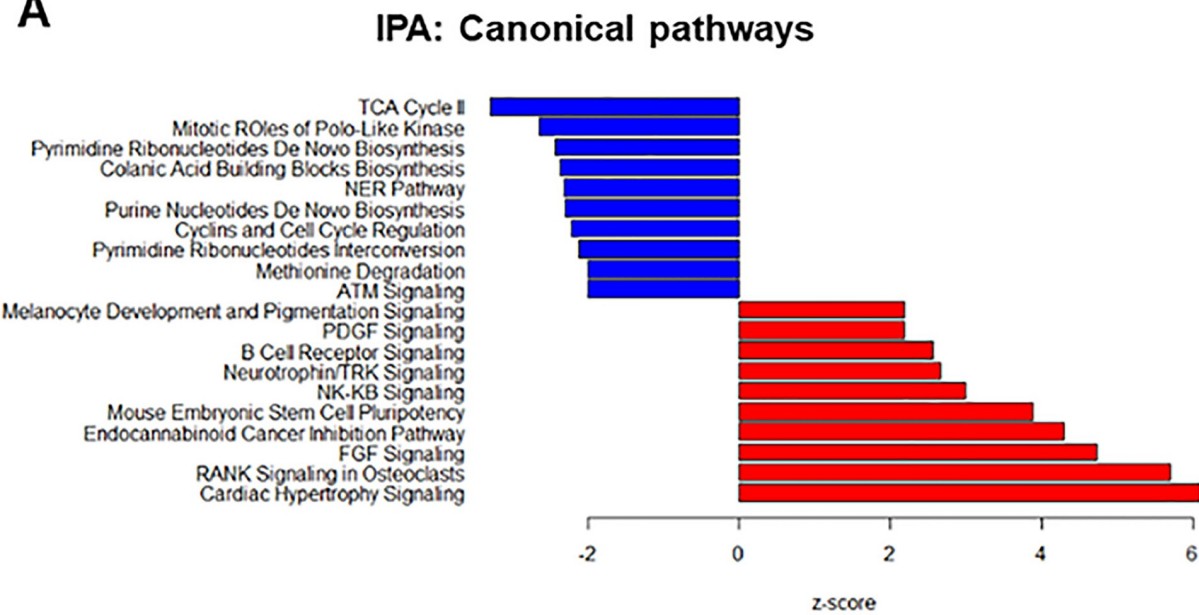

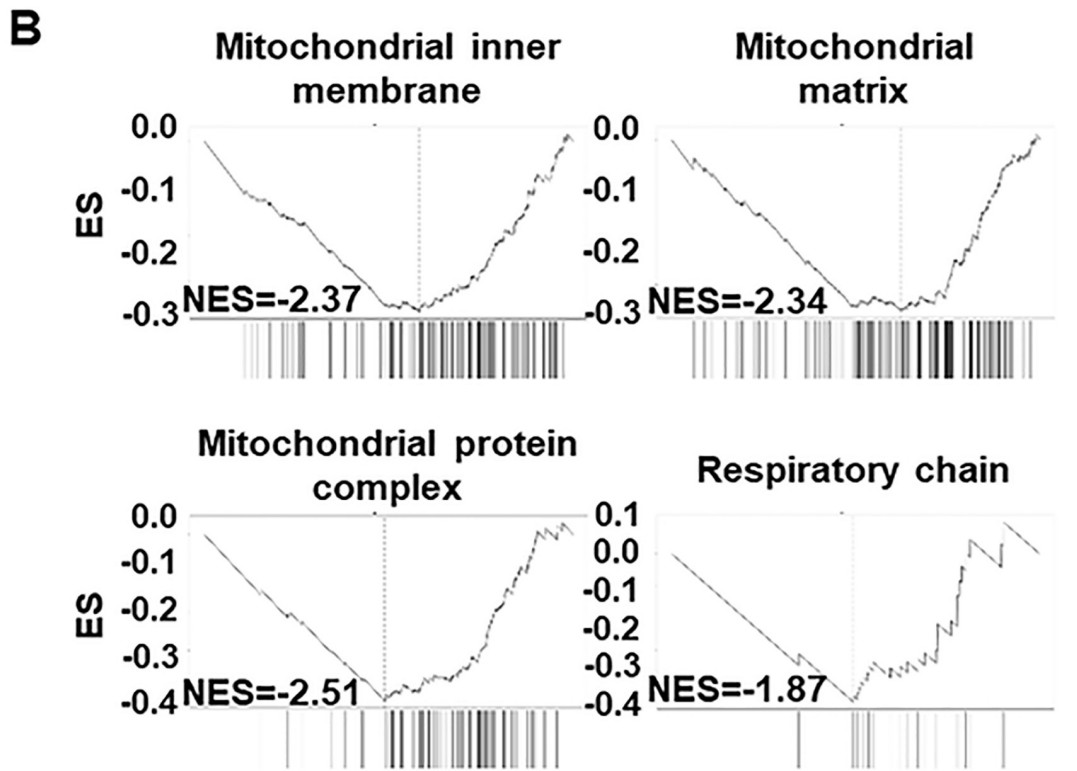

**Fig 3. RNA-seq data indicate that EtOH may alter mitochondria-associated gene expression.** EPC1, EPC2 and EPC3 cells were treated with or without EtOH (2%, 8 h) and subjected to RNA-Seq and data analysis. **A**. Bar diagram of canonical pathways significantly dysregulated by EtOH (p<0.01, z<-2 and >2). **B**. GSEA suggested negative enrichment of mitochondria-associated gene sets. ES, Enrichment score; NES, Normalized enrichment score.

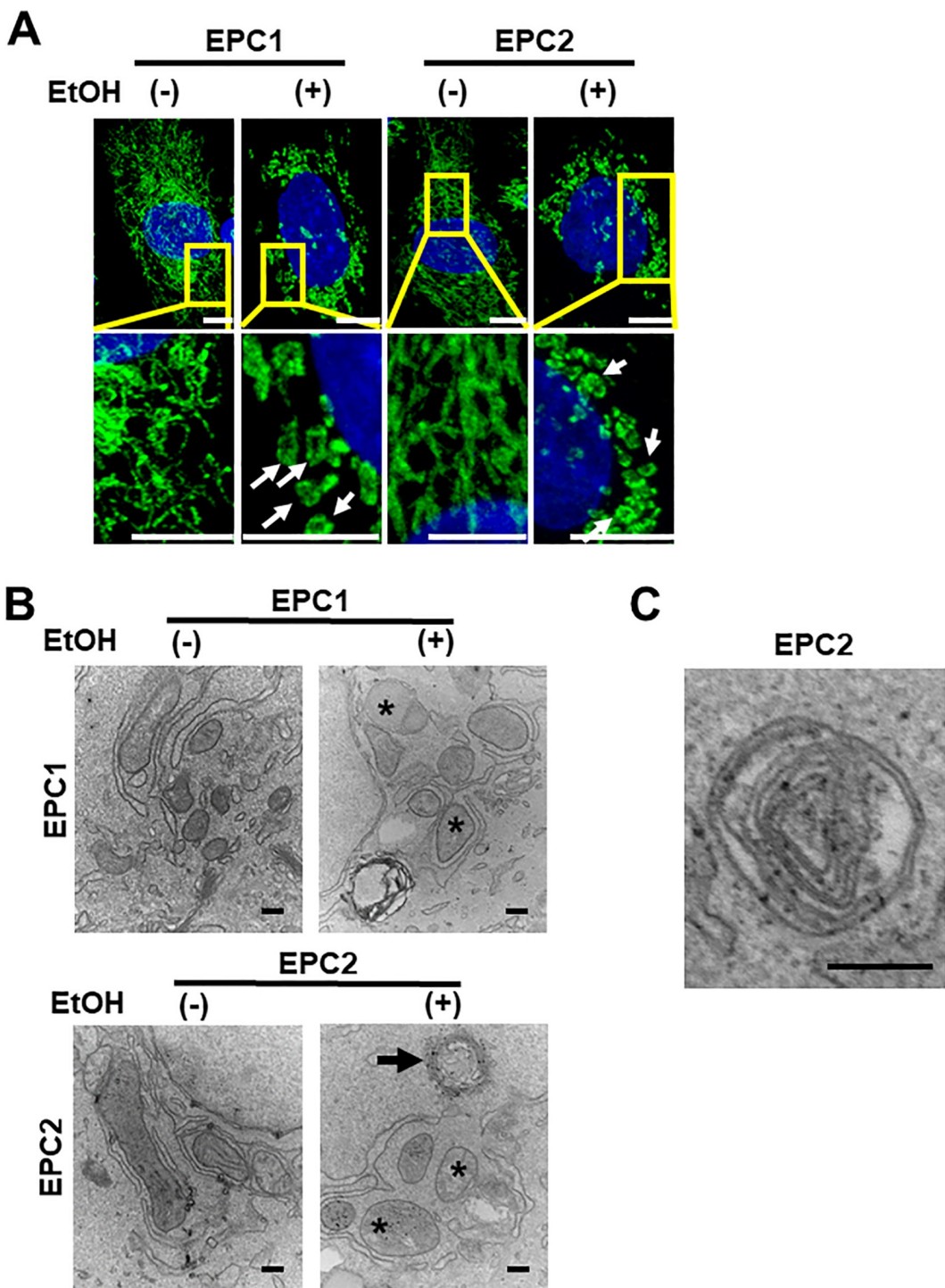

**Fig 4. EtOH affects mitochondrial structure.** EPC1 and EPC2 cells were treated with or without EtOH (2%, 8 h). **A.** Immunofluorescence and confocal microscopy visualized ATPB-stained mitochondrial architecture. Inset denotes the areas enlarged to demonstrate normal filamentous mitochondria in EtOH-untreated cells and ring- or donut-shaped abnormal mitochondria (white arrows) in EtOH-treated cells. Original magnification, 63X. Scale bar, 5 μm. **B and C**. TEM captured ultrastructural changes. *, enlarged and swollen mitochondria with few or no cristae in **B**. A double-membrane vesicle with contents reminiscent of an autophagic vesicle in **C**. Original magnification: 60,000X; Scale bars, 200 nm in **B** and **C**.

depolarization occurs as an early event associated with mitochondrial dysfunction, especially in dying cells [31]. EtOH exposure induces mitochondrial depolarization in murine hepatocytes [32]. We performed flow cytometry to evaluate how EtOH may influence $\Delta\Psi$m in esophageal keratinocytes. To this end, EtOH-exposed cells were stained concurrently with $\Delta\Psi$m-sensitive MitoTracker® Deep Red (MTDR) and $\Delta\Psi$m-insensitive MitoTracker® Green (MTG) dyes to determine $\Delta\Psi$m and total mitochondrial mass, respectively. In monolayer culture, 2% EtOH induced a small subset (<4%) of EPC2 cells with decreased MTDR expression (i.e. $\Delta\Psi$m) in a transient manner with its peak around 8 h (Fig 5A). Within the cell population with decreased $\Delta\Psi$m, the MTG signal intensity was stable over time (Fig 5A), in agreement with the expected function of the respective MitoTracker® dyes in our experimental system. The cell population with decreased $\Delta\Psi$m was also induced in EPC1 cells after 8 hours of 2% EtOH exposure (Fig 5B). Induction of the $\Delta\Psi$m-decreased cell population was recapitulated in 2% EtOH-exposed EPC2 cells in 3D organoids, although the change (< 2-fold) was smaller than that in monolayer culture conditions (Fig 5C). We further analyzed murine esophageal keratinocytes isolated from mice following treatment with 31.5% EtOH via oral gavage. Similar to EPC2-derived 3D organoids, esophagi from EtOH-treated mice contained a subset of cells with decreased $\Delta\Psi$m (Fig 5D). Due to a reactive chemical property, the MitoTracker® dyes may be permanently retained in depolarized mitochondria, leading to potential misinterpretation of cells with dysfunctional mitochondria as intact [33], and thereby underestimating the number of cells with mitochondrial dysfunction. Nevertheless, our data suggest that EtOH may induce mitochondrial depolarization at least in a small subset of esophageal keratinocytes.

Mitochondria play a central role in ATP generation via oxidative phosphorylation (OXPHOS). Cellular stresses may impact mitochondrial ATP production. As a readout for mitochondrial functions, we evaluated the ATP level in EtOH-treated EPC2 cells. We observed a significant decrease in the ATP level upon EtOH exposure (2%, 8 h) (Fig 6A), suggesting that EtOH exposure may induce mitochondrial dysfunction in EPC2 cells.

We suspected that the OXPHOS pathway is impaired under EtOH-induced stress. Indeed, liquid chromatography mass spectrometry (LC-MS) demonstrated a significant downregulation of TCA cycle-related metabolites in EtOH-treated EPC2 cells (Fig 6B). The amount of acetyl-CoA and citrate/isocitrate was increased in EtOH-treated cells, suggesting inhibition in the first step of the TCA cycle. Consistent with this, several TCA metabolites were greatly decreased with EtOH exposure (Fig 6B), indicating decreased flux through oxidative phosphorylation pathways and correlating with the observed EtOH-induced mitochondrial dysfunction.

To confirm this, we next investigated the effect of EtOH on mitochondrial respiration by high resolution respirometry. While EtOH treatment did not affect the basal oxygen consumption rate, we observed a significant decrease in substrate specific response of the mitochondrial complexes of the ETC (Fig 6C), where EtOH strongly inhibited the activities of complexes I, II, III and IV in the presence of their respective substrates.

Impaired ETC function may contribute to oxidative stress through the generation of free radicals and reactive oxygen species [34]. We therefore analyzed the cellular redox state by LC-MS. The GSH:GSSG ratio (reduced glutathione vs. oxidized glutathione), an indicator of oxidative stress, was found to be decreased in EtOH-treated cells (Fig 6D). In agreement, EtOH induced generation of mitochondrial superoxide, as indicated by flow cytometry for MitoSOX, a mitochondria-targeted reporter fluorescent dye for superoxide. Furthermore, this superoxide induction was suppressed by MitoCP, a mitochondria-targeted superoxide antioxidant (Fig 6E), suggesting a role for mitochondrial superoxide in EtOH-induced oxidative stress.

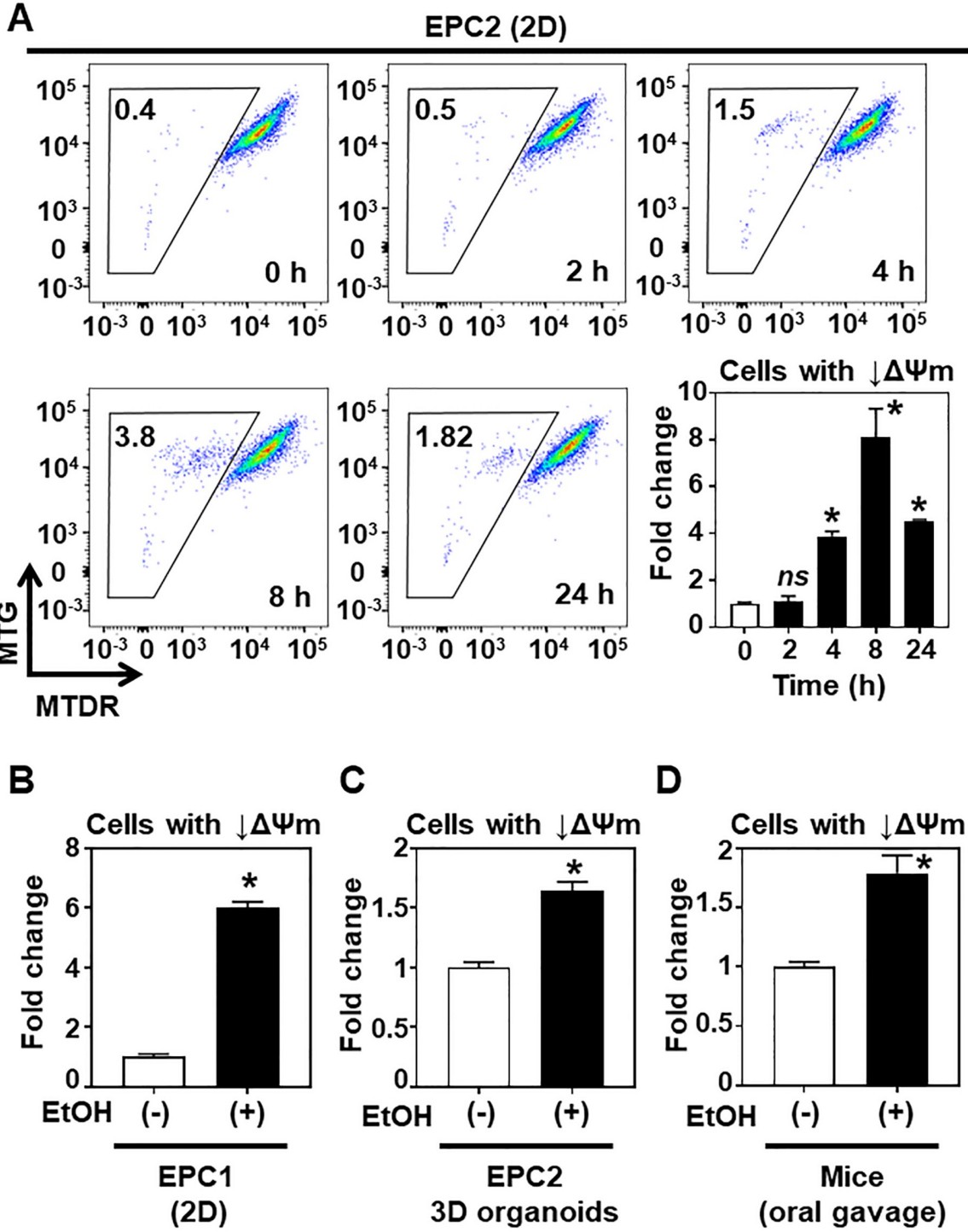

**Fig 5. EtOH decreased ΔΨm in a subset of esophageal keratinocytes.** EtOH-exposed cells were stained with MTG and MTDR for flow cytometry to determine mitochondrial mass and membrane potential (ΔΨm), respectively. **A-C**. Indicated cell lines were treated with 2% EtOH or untreated [EtOH (-), control] in monolayer (2D) and 3D organoid culture conditions. Representative scatter plots for MTG and MTDR at indicated time points with bar diagram (mean ± sem, n = 3 per condition) depicting quantification of cells showing reduced ΔΨm in **A**. Quantification of cells with reduced ΔΨm for EPC1 cells in 2D and EPC2 cells in 3D organoids are shown in bar diagrams in **B and C**, respectively. **D**. Following oral gavage with 31.5% EtOH (5 g/kg in PBS) or PBS [EtOH (-), control], mice were sacrificed 6 hours later to isolate esophageal epithelial cells for MTG and MTDR staining. Bar diagram depicts quantification of cells showing reduced ΔΨm from the murine esophagi. n = 3 per condition. *p<0.05 vs. time 0 or EtOH (-); ns, not significant vs. time 0 or EtOH (-), using student's t-test.

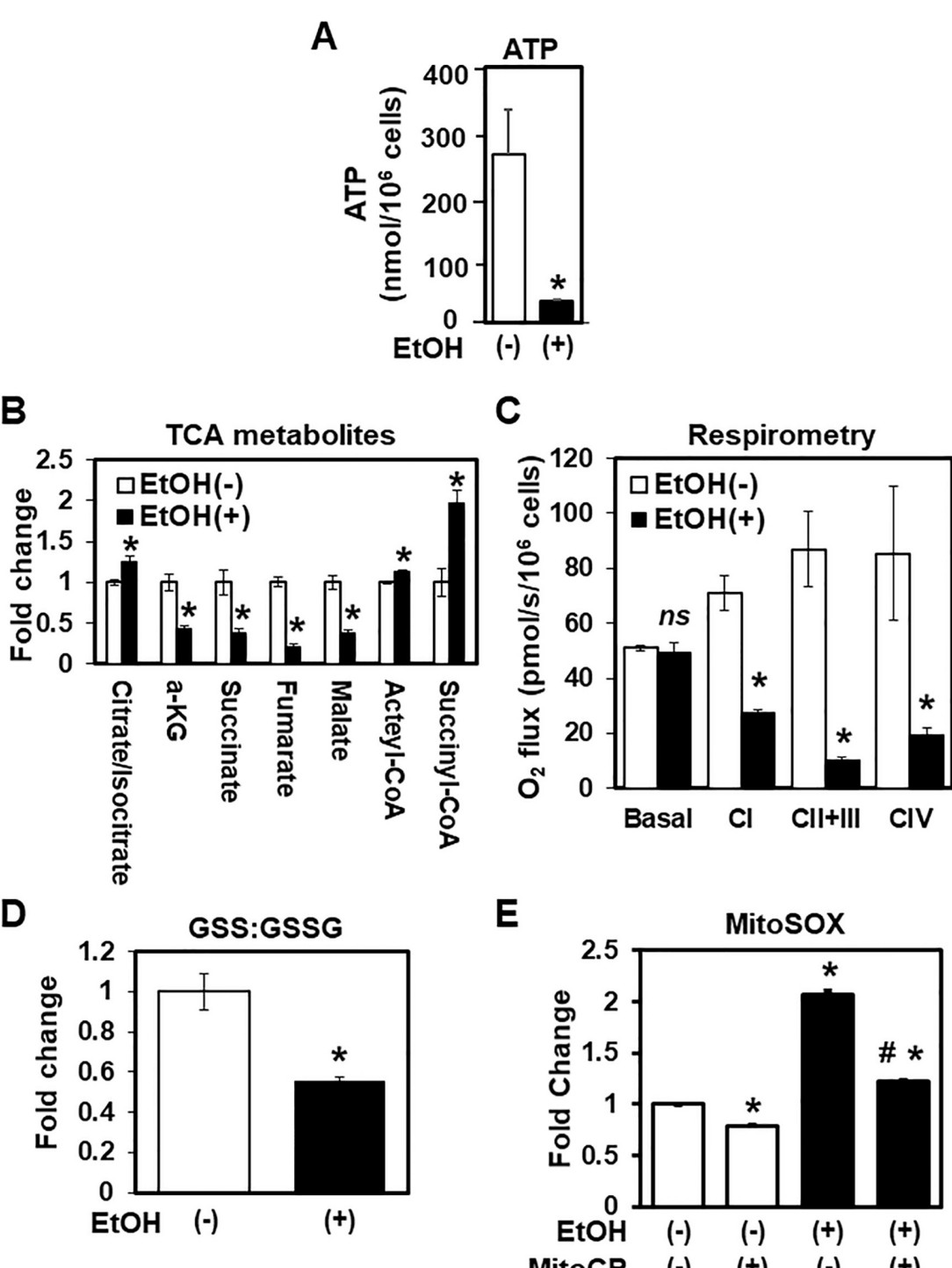

**Fig 6. EtOH reduces cellular ATP content and ETC activity to promote oxidative stress.** EPC2 cells were treated in triplicate with or without EtOH (2%, 8 h) to evaluate mitochondrial functions. Data present mean ± sem. n = 3 per condition in **A-E**. **A**. Total cellular ATP. *, p<0.05 vs. EtOH (-). **B**. LC-MS determined TCA cycle metabolites.*, p<0.05 vs. EtOH (-); **, p<0.01 vs. EtOH (-). **C**. High resolution respirometry was performed to determine mitochondrial respiration. Bar diagram demonstrates decreased substrate-dependent oxygen consumption under EtOH exposure. *, p<0.05 and ns, not significant, vs. EtOH (-) in the basal O₂ flux. **D**. LC-MS determined reduced and oxidized forms of glutathione (GSH and GSSG). Bar diagram shows the GSH: GSSG ratio. *, p<0.05 vs. EtOH (-). **E**. Flow cytometry for MitoSOX to determine mitochondrial superoxide. Cells were exposed to EtOH in the presence or absence of Mito-CP. *, p<0.05 vs. EtOH (-) and Mito-CP (-); #, p<0.05 vs. EtOH (+) and Mito-CP (-).

To further investigate how EtOH may induce mitochondrial dysfunction, we performed RNAi experiments directed against CYP2E1 and ADH1B, two major alcohol metabolizing enzymes. In EPC2 cells, partial knockdown of CYP2E1 and ADH1B appeared to prevent EtOH from inducing cells with ΔΨm reduction (S6 Fig), suggesting that EtOH oxidation mediated by these enzymes may contribute to mitochondrial dysfunction in esophageal keratinocytes.

## Autophagy may provide cytoprotection against EtOH-induced mitochondrial dysfunction and oxidative stress in esophageal keratinocytes

The majority of EPC1 and EPC2 cells remained viable in the presence of 2% EtOH for at least 8 h (Fig 2B) despite the observed mitochondrial dysfunction and oxidative stress during this time period, suggesting that cells may activate compensatory cytoprotective mechanisms to promote survival. Based on our prior work studying esophageal cell response to acetaldehyde, the chief metabolite of EtOH [5], we suspected that autophagy may occur in esophageal keratinocytes exposed to EtOH. We performed immunoblot analysis for LC3B and SQSTM1/p62 (p62) (Fig 7A) to evaluate autophagic flux in EPC1 and EPC2 cells. As a key structural component of the autophagic vesicles (AV), LC3B-I is rapidly converted to LC3B-II, the lipidated form of LC3B protein, during AV formation. The p62 protein brings cargo into AV where both the cargo and p62 undergo degradation upon the fusion of AV with lysosomes. Thus, p62 downregulation is indicative of increased autophagy flux. When cells were treated with EtOH, LC3B-II was increased in response to EtOH exposure (Fig 7A and S6A Fig), suggesting increased AV formation. Concurrently, p62 level was decreased following EtOH exposure (Fig 7A and S6B Fig). Pharmacologic inhibition of autophagy flux by chloroquine (CQ) increased both LC3B-II and p62 levels, inhibiting both basal autophagic flux as well as the increased flux induced by EtOH exposure (Fig 7A, S6A and S6B Fig). We also evaluated AV number in EPC1 and EPC2 cells by flow cytometry using Cyto-ID, a fluorescent probe for AV. EtOH exposure increased the Cyto-ID signal, consistent with upregulation of autophagy. Cyto-ID staining was further increased by CQ (Fig 7B and 7C), indicating accumulation of AV due to impaired AV clearance with CQ treatment.

We next evaluated how pharmacologic inhibition of autophagic flux by CQ may influence EtOH-induced oxidative stress, mitochondrial ΔΨm, and cell viability in EPC1 and EPC2 cells. In response to 2% EtOH exposure (8 h), both cell lines showed increased mitochondrial superoxide level (Fig 8A and 8B). and development of cellular subsets with decreased ΔΨm (Fig 8C and 8D). 2% EtOH alone did not affect EPC1 cell viability at 8 h time point as evaluated by WST-1 assays (Fig 8E). Viable EPC2 cells were decreased by 10% upon EtOH exposure (Fig 8F) although flow cytometry for DAPI-stained cells did not detect a significant increase in dead cells at this time point (Fig 2B), reflecting probably differences in the assay principles and sensitivity. CQ alone increased the cellular subset with decreased ΔΨm without affecting basal superoxide expression or cell viability. In the presence of 2% EtOH, CQ augmented mitochondrial superoxide (Fig 8A and 8B), as well as accumulation of the cell population with decreased ΔΨm (Fig 8C and 8D), while reducing cell viability significantly (Fig 8E and 8F). These findings suggest association between mitochondrial dysfunction and the cytoprotective role of autophagy under EtOH-induced stress.

## The mTORC1 pathway may be suppressed in EtOH-treated esophageal keratinocytes

We explored pathways that may account for the gene expression changes observed under EtOH-induced stress. We utilized the IPA software to analyze our RNA-seq data (GSE143479,

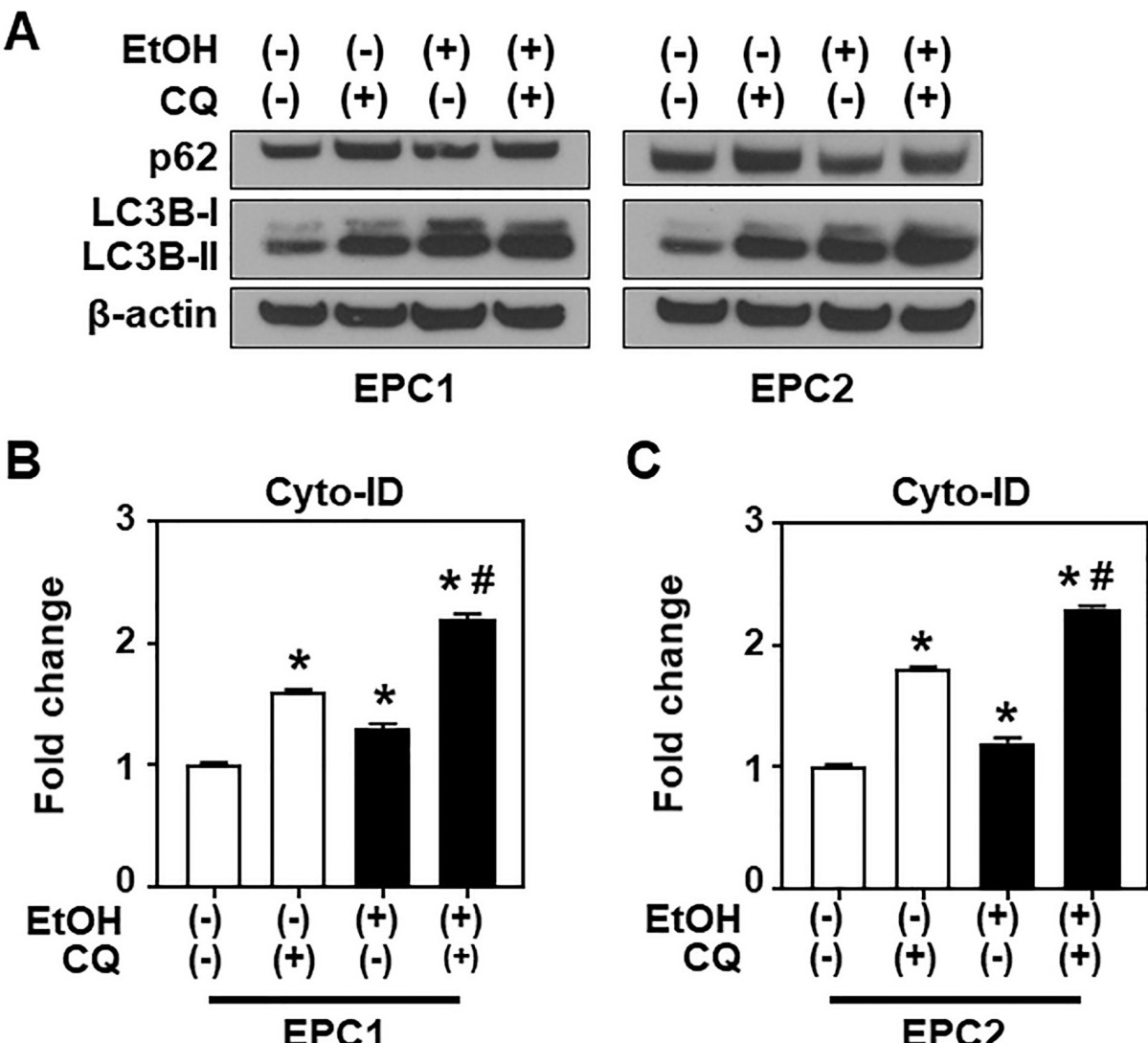

**Fig 7. EtOH induces autophagy flux.** EPC1 and EPC2 cells were exposed in triplicate to EtOH (2%, 8 h) in the presence or absence of 1 μg/ml CQ, an autophagy flux inhibitor. **A**. Immunoblot analysis for indicated autophagy mediators with β-actin serving as a loading control. Densitometry of LC3B-II and p62 levels relative to β-actin was shown in S6A and S6B Fig. Immunoblots represent at least two independent experiments with similar results. **B and C**. Flow cytometry for Cyto-ID to determine AV. Bar diagrams show relative signal intensity compared to that for EtOH (-) CQ (-), which is defined as 1. Data represent mean ± sem. n = 3 per condition*, p<0.05 vs. EtOH (-) and CQ (-); #, p<0.05 vs. EtOH (+) and CQ (-), using student's t-test.

Fig 3) for the cascade of upstream transcriptional regulators. The mTOR and its network, which negatively regulates autophagy and induces several anabolic pathways, was suggested to be suppressed in EtOH-treated cells (Fig 9A and 9B). To confirm the downregulation of this pathway following EtOH exposure, we performed immunoblotting for mTORC1 targets. Following EtOH exposure, we observed robust inhibition of phosphorylation of the mTORC1 target S6 in both EPC1 and EPC2 cells (Fig 9C and S6C Fig), as well as decreased phosphorylation of the mTORC1 target 4EBP-1 in EPC2, but little if any in EPC1 cells (Fig 9C and S6D Fig). This likely reflects some differences in the mTORC1 downstream pathways between the two cell lines.

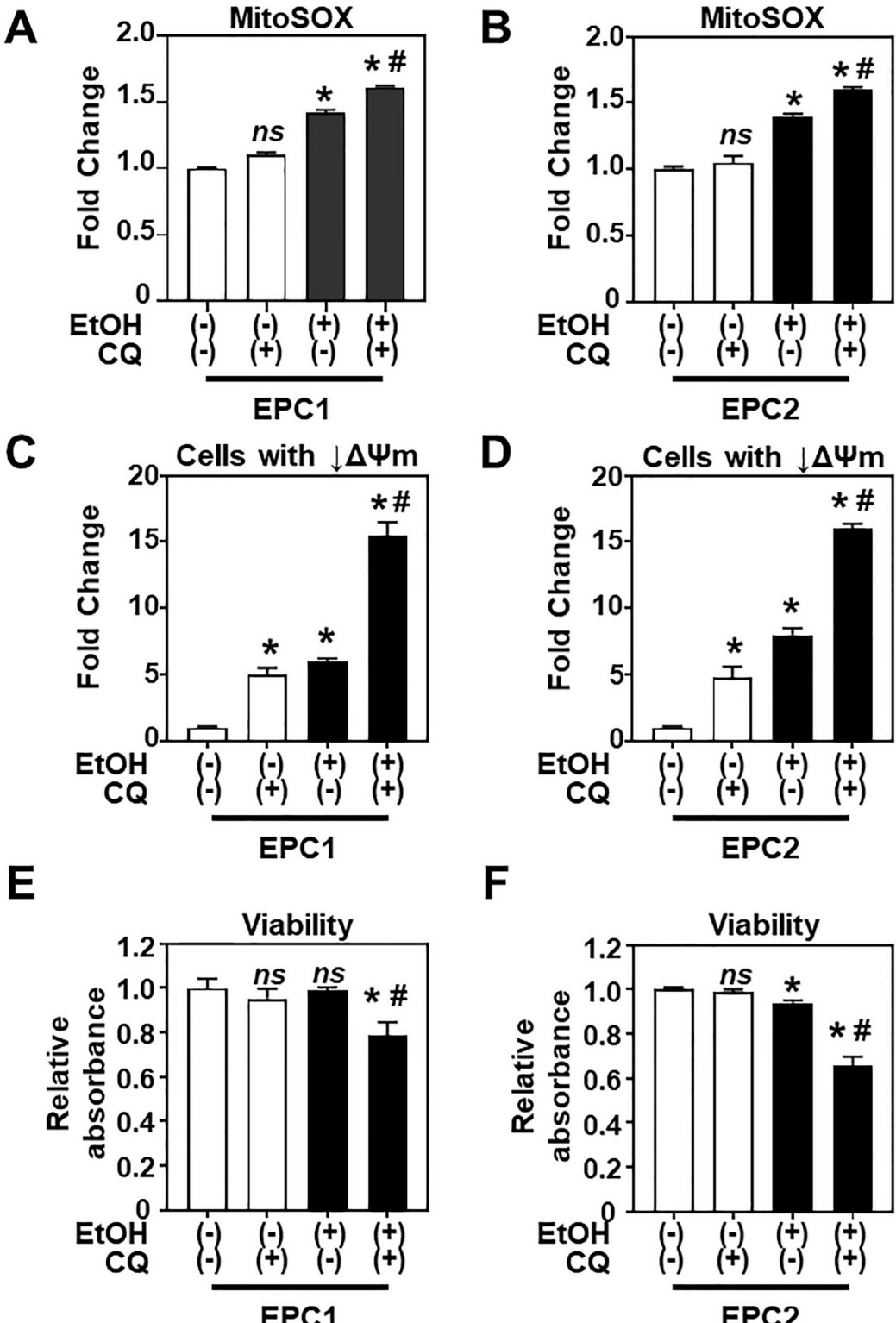

**Fig 8. Inhibition of autophagic flux increases EtOH-induced oxidative stress, induces mitochondrial depolarization, and decreases cell viability.** EPC1 and EPC2 cells were exposed to EtOH (2%, 8 h) in the presence or absence of 1 μg/ml CQ. **A and B**. Flow cytometry for mitochondrial superoxide (MitoSOX). **C and D**. Flow cytometry for mitochondrial ΔΨm (MTDR) and mass (MTG). **E and F**. WST-1 assays to determine cell viability. Bar diagrams show fold-difference in **A-D** and relative absorbance in **E and F**, compared to EtOH (-) CQ (-) as 1. Data represent mean ± sem. n = 3 per condition in **A-D**; n = 4 in **E and F**. *, p<0.05 vs. EtOH (-) and CQ (-); *ns*, not significant vs. EtOH (-) and CQ (-); #, p<0.05 vs. EtOH (+) and CQ (-), using student's t-test.

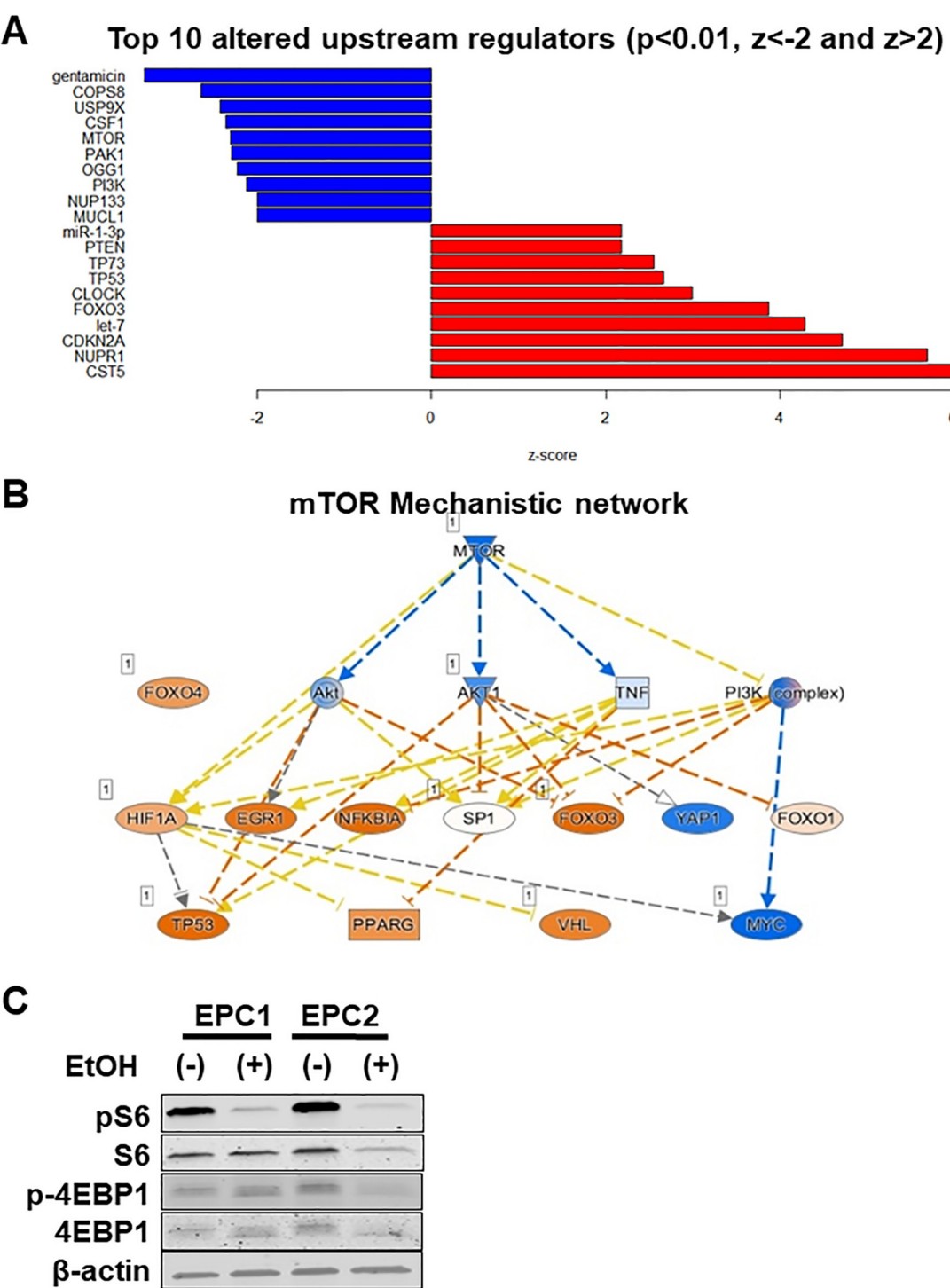

**Fig 9. Pathway analysis suggests altered mTORC1 signaling in EtOH-treated EPCs.** IPA analysis of RNA-seq data of EPC1, EPC2 and EPC3 cells exposed to 2% EtOH. **A**. Most activated (red) and inhibited (blue) upstream regulators (top 10 and bottom 10) (cut-off: q<0.01, z<-2 and Z>2). **B**. The mechanistic network of upstream regulator mTOR demonstrates nodes activated (orange) or inhibited (blue). Color intensity represents level of activation or inhibition with darker shades meaning strong activation or inhibition. Dashed lines indicate indirect inhibition (orange), activation (blue), or predictions inconsistent with the state of downstream molecule (yellow). Dashed grey lines represent unpredicted interactions. Numbers in boxes indicate the number of isoforms for the indicated gene. **C**. EPC1 and EPC2 cells were treated with or without EtOH (2%, 8 h) for immunoblot analysis to determine phosphorylation of mTORC1 substrates S6 protein and 4E-BP1, compared to total S6 and 4E-BP1 proteins, with β-actin serving as a loading control. Densitometry determined phosphorylation levels of each protein relative to total protein is shown in S6C and S6D Fig. Immunoblots represent one of two independent experiments with similar results.

## Autophagy influences AMPK activation in the setting of EtOH-induced mitochondrial dysfunction in EPC2 cells

We investigated further how EtOH-induced stress may suppress mTORC1 signaling. AMPK is a major upstream inhibitor of mTORC1 [9], and serves as a master energy sensor to maintain cellular energy homeostasis. It is activated when cellular ATP is decreased with a reciprocal increase in AMP. EPC2 cells displayed increased AMPKα Thr172 phosphorylation in response to EtOH exposure (Fig 10A and S6E Fig), suggesting that AMPK is activated in response to EtOH. Unexpectedly, basal AMPK phosphorylation was readily detectable in EPC1 cells without EtOH stimulation, and EtOH did not further augment AMPK phosphorylation (Fig 10A and S6E Fig). EtOH-induced AMPK phosphorylation was augmented in EPC2, but not in EPC1, when autophagy flux was blocked by CQ, suggesting that inhibition of autophagy may exacerbate the intracellular energy crisis and mitochondrial dysfunction induced by alcohol (i.e. decreased ATP production, increased MitoSOX, decreased membrane potential) through an AMPK-related pathway in EPC2, but not EPC1. We suspected that AMPK activation may alleviate mitochondrial dysfunction in EPC2 cells. Pharmacologic AMPK activation by AICAR reduced the number of EPC2 cells displaying mitochondrial depolarization after exposure to EtOH (Fig 10B). Additionally, AICAR partially attenuated the EtOH-induced ATP depletion observed in EPC2 cells (Fig 10C). These findings suggest that AMPK may have a protective role in cellular energetic and mitochondrial functional homeostasis under EtOH-induced stress.

## Discussion

Due to its physiologic location, the esophageal epithelium is vulnerable to high concentrations of EtOH exposure associated with alcohol consumption, however to date there has been little evidence demonstrating the effects of alcohol on esophageal epithelial biology and homeostasis. In this study, we have demonstrated for the first time that EtOH induces mitochondrial dysfunction and oxidative stress in esophageal keratinocytes where autophagy may provide cytoprotection against EtOH-induced cell death as validated in two independent cell lines EPC1 and EPC2. Additionally, the AMPK-mTORC1-autophagy axis may regulate mitochondrial homeostasis under EtOH-induced oxidative stress (Fig 11). However, this model was supported by data with EPC2 cells only. Notably, EtOH barely influenced phosphorylation of AMPK or 4EBP1 in EPC1 cells despite suppression of S6 phosphorylation (Figs 9C and 10A). Thus, autophagy may be activated via a mechanism independent of the AMPK-mTORC1 pathway in EPC1 cells with EtOH-induced mitochondrial dysfunction. Alternatively, relatively high basal AMPK phosphorylation in EPC1 (Fig 10A) may suggest a preexisting vulnerability in these cells in the utilized cell culture conditions (e.g. nutrient deficiency in cell culture), potentially masking an effect of EtOH on the AMPK-mTORC1 axis in EPC1. All EPC cell lines utilized in this study were originally immortalized by human telomerase hTERT. A recent study shows that hTERT may have a non-canonical mitochondrial function to inhibit AMPK phosphorylation and autophagy activation under oxidative stress [35], raising the possibility that potentially variable hTERT expression level may differentially influence AMPK, downstream mTOR pathway, and autophagy in EPC cell lines. Future investigations should include non-immortalized primary esophageal keratinocytes for functional validation of the role of the AMPK-mTORC1 axis (Figs 9 and 10) as well as detailed evaluation of mitochondrial function (Figs 5 and 6). Additionally, while our data suggest that mitochondria depolarization may occur in mice receiving oral EtOH gavage (Fig 5D), further *in vivo* studies investigating this pathway are currently lacking. Further investigation is necessary to validate the model utilizing multiple cell lines, genetically engineered mice, and human subjects. Nevertheless, this study

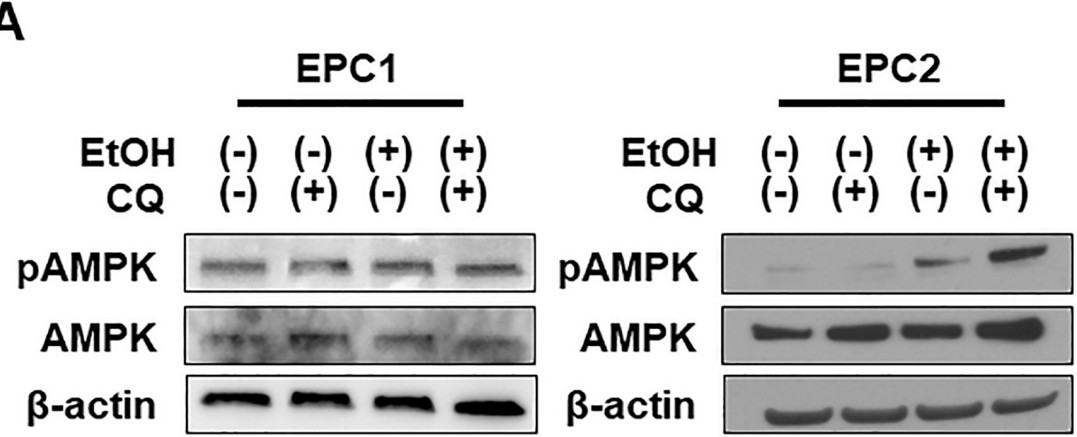

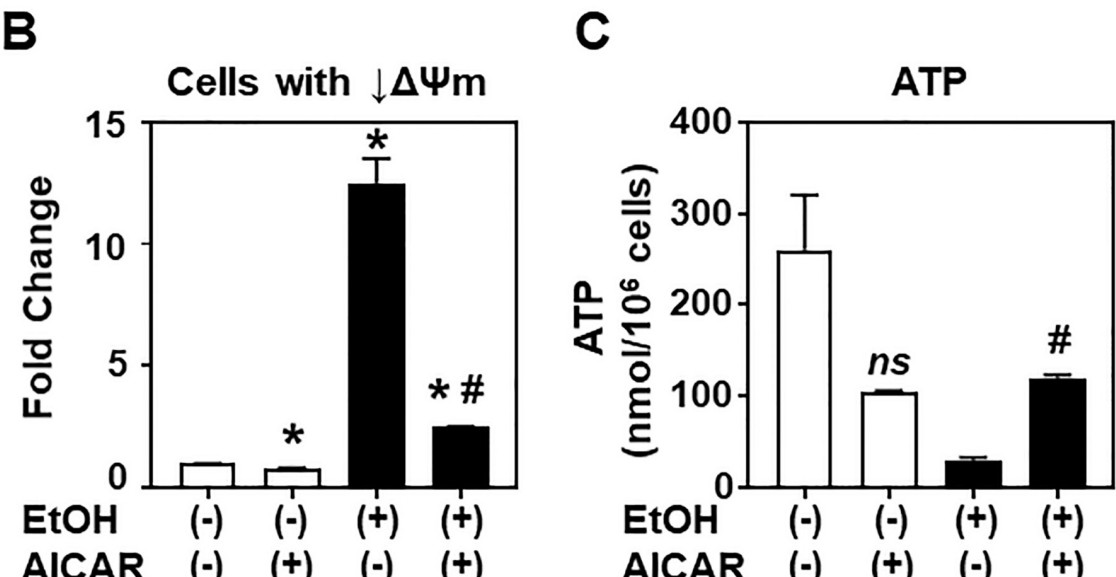

**Fig 10. EtOH may activate AMPK in concert with autophagy to affect mitochondrial function in EPC2 cells.** Cells were treated with or without EtOH (2%, 8h) in the presence or absence of 1 μg/ml CQ or AMPK activator AICAR. **A**. EPC1 and EPC2 cells were subjected to immunoblot analysis for AMPKα phosphorylation, comparing to total AMPKα protein with β-actin serving as a loading control. Densitometry-determined phosphorylation level of AMPKα relative to total protein is shown in S6 Fig. Immunoblots represent one of two independent experiments with similar results. **B**. Flow cytometry for mitochondrial ΔΨm (MTDR) and mass (MTG). Bar diagrams show fold-difference, compared to EtOH (-) AICAR (-), which was defined as 1. Data represent mean ± sem. n = 3 per condition. *, $p < 0.05$ vs. EtOH (-) AICAR (-); #, $p < 0.05$ vs. EtOH (+) AICAR (-), using student's t-test. **C**. Cellular ATP levels in EtOH-treated cells in the presence of AICAR. *, $p < 0.05$ vs. EtOH (+)AICAR (-); *ns*, not significant vs. EtOH (-) AICAR (-), n = 3 per condition.

provides novel insights about how EtOH may affect esophageal keratinocyte energy homeostasis as discussed below.

EtOH and acetaldehyde in circulation have been associated with mitochondrial damage in the liver, the brain, the heart and muscle [32, 36]. We find that esophageal keratinocytes may tolerate continuous exposure to 2% (350 mM) EtOH, which is much higher than most other cell types do. However, we observed variability in EtOH sensitivity amongst the cell lines used in this study, likely reflecting their genetic background, as corroborated by PCA (S4 Fig),

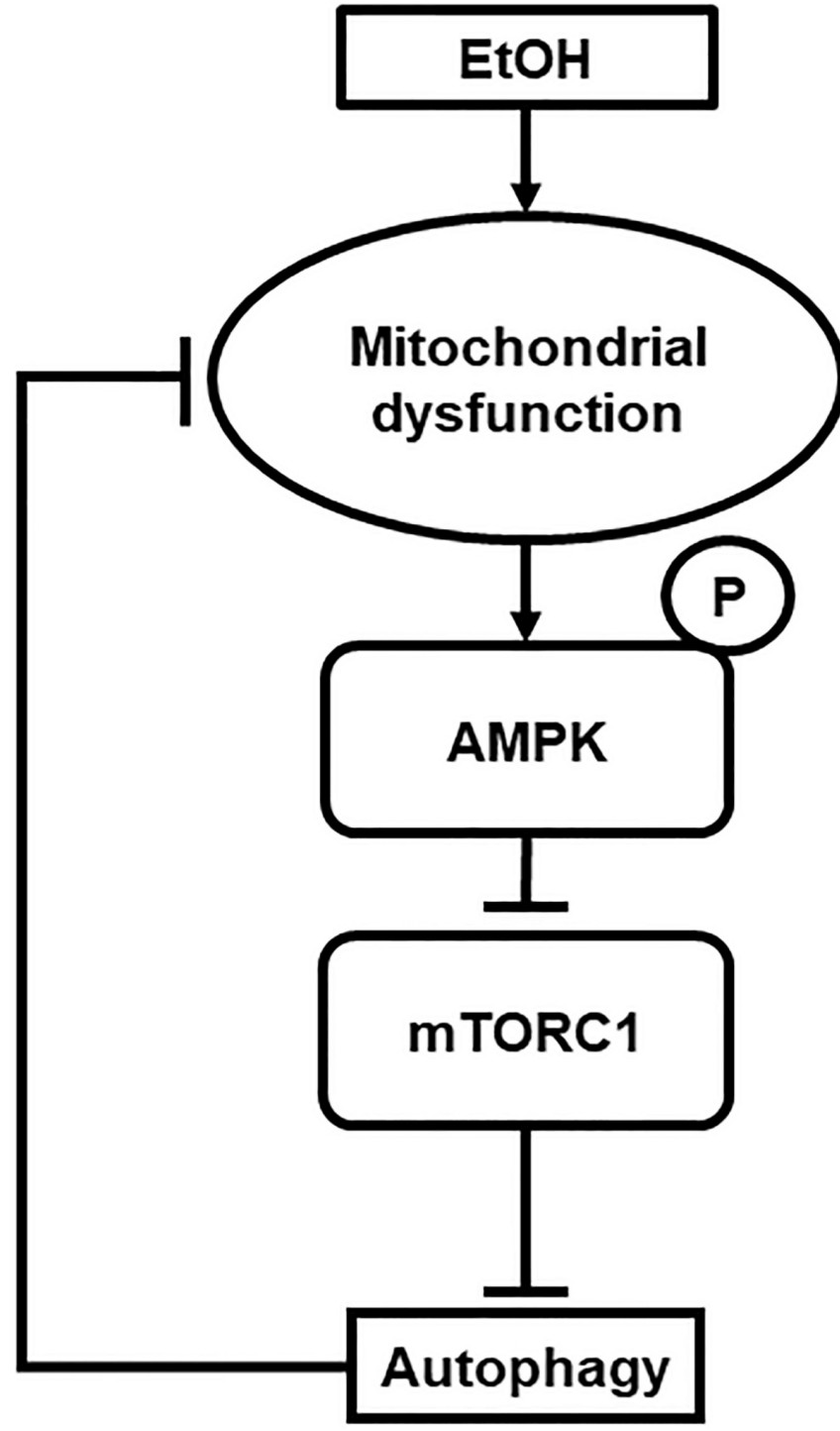

**Fig 11. Model.** EtOH may cause mitochondrial dysfunction to promote energetic crisis, resulting from decreased ATP production and oxidative stress. ATP depletion may result in AMPK activation leading to mTORC1 suppression, which in turn may activate autophagy to restore mitochondrial homeostasis, cellular energetics, and redox balance. Note that this model was supported by data from EPC2, but not EPC1 in this study.

where overall gene expression pattern was closer in EPC1 and EPC2, as compared with EPC3. We have previously demonstrated that acetaldehyde causes oxidative stress and DNA damage in murine esophageal keratinocytes, which was augmented in cells with loss of Aldh2 [3, 5], a mitochondrial enzyme essential for acetaldehyde clearance. Single nucleotide polymorphisms resulting in mutant ALDH2 causes delayed acetaldehyde clearance and enhanced EtOH toxicity. Such ALDH2 mutations are common in East Asians and linked to esophageal cancer susceptibility in alcohol users [37]. Interestingly, EPC1 and EPC2 cells carry wild-type *ALDH2* while EPC3 cells, which were isolated from a Japanese patient, carry the mutant form of *ALDH2* (unpublished findings by SO, personal communications). It is therefore tempting to speculate that this *ALDH2* mutation may contribute to increased death of EPC3 cells upon EtOH exposure (Fig 2B).

RNA-seq results suggested that EtOH exposure results in mitochondrial dysfunction in esophageal keratinocytes (Fig 3). These gene expression changes were corroborated by several findings that indicated EtOH-induced mitochondrial dysregulation, including decreased ATPB protein expression (Fig 4A), swollen mitochondria with cristae changes (Fig 4B), decreased cellular ATP level, suppression of TCA metabolites and respiration, and increased levels of mitochondrial superoxide and cellular oxidative stress (Fig 6). Moreover, we have detected a small subset of cells with decreased $\Delta\Psi$m, indicating mitochondrial depolarization, induced by alcohol treatment (Fig 5). These cells may represent early apoptotic cells (S2 Fig), as EtOH significantly decreased cell viability in the presence of CQ, along with a reciprocal increase in the fraction of cells with mitochondrial depolarization and superoxide levels (Fig 8). Assessment of $\Delta\Psi$m coupled with RNAi also offered some insight into how EtOH may induce mitochondrial dysfunction. Besides ALDH2, our data suggest that the alcohol metabolizing enzymes ADH1B and CYP2E1 may contribute to mitochondrial dysfunction through EtOH oxidation and acetaldehyde production (S5 Fig). Consistent with this, CYP2E1 has been implicated in ROS generation and mitochondrial dysfunction in alcohol-induced hepatic steatosis [38–41]. Additionally, our RNAi experiments show for the first time that esophageal keratinocytes can metabolize EtOH via ADH1B and CYP2E1 in vitro without requiring hepatic metabolism. As suggested in epidermal keratinocytes [42, 43], alcohol tolerance may represent a unique attribute of keratinocytes that comprise the stratified squamous epithelium to form the barrier between the body and the outside world. Given exposure to high concentration (>5%) of EtOH during alcohol drinking, our $\Delta\Psi$m data in mice (Fig 5D) confirms the impact of EtOH upon esophageal epithelial cells *in vivo*. The relatively modest change in the $\Delta\Psi$m seen in 3D organoids and mice (Fig 5C and 5D), as compared to monolayer culture conditions (Fig 5A and 5B), may be accounted for by a protective effect of the epithelial barrier or extracellular matrix present in 3D organoids or tissues, which may serve to limit EtOH exposure, even at higher concentrations. In agreement, the extent of suppression of cell viability by 2% EtOH was lower in 3D organoids (<50%; Fig 1C) as compared to monolayer culture conditions (>75%; Fig 2A).

It is well established that EtOH treatment results in conditions that promote mitochondrial ROS formation, as corroborated by our data (Fig 8A and 8B). ROS have been shown to form mitochondrial (mt) DNA adducts and mtDNA damage upon EtOH exposure [44]. Additionally, EtOH affects the mitochondrial antioxidant mechanism, further exacerbating this damage [45]. ROS serve key roles in cellular signaling to maintain homoeostasis; however, high levels of ROS promote widespread cellular injury and DNA damage, a precursor for malignant transformation [46]. Perturbations in mitochondrial respiration result in dramatic increases in cellular ROS levels [47]. Our RNA-seq results suggest that DNA and mitochondria may be major targets of EtOH-induced cell injury in esophageal keratinocytes (Fig 3). We find that mitochondrial superoxide may represent EtOH-induced cellular ROS associated with concurrent

mitochondrial respiratory failure, metabolic stress, and decreased cellular ATP levels (Fig 6). Through these mechanisms, mitochondrial superoxide may promote DNA damage in EtOH-exposed keratinocytes.

Autophagy limits alcohol-induced hepatotoxicity and steatosis in mice [6, 7]. We have previously demonstrated that autophagy prevents acetaldehyde from inducing DNA damage and apoptosis in esophageal keratinocytes [5]. Using TEM we found that alcohol induced intracellular vacuolar structures (Fig 3B) that likely represent involutional mitochondria and some autophagic vesicles (Fig 3C). In our analysis, the number of autophagic vesicles may have been underestimated due to the apparent lack of double membrane, which could be accounted for by insufficient resolution to detect distended double membranes in vacuolized and distended organelles. We have previously documented mitophagy in transformed EPC2 derivative cells under oxidative stress stimulated by transforming growth factor-β [18]. It is plausible that autophagy may have a role in removal of EtOH-induced dysfunctional mitochondria via mitophagy to promote cell survival under ethanol-induced stress, warranting further study.

Multiple pathways have been implicated in the regulation of autophagy within the context of alcohol-induced hepatic steatosis [48]. Consistent with this, several of the upstream regulators of autophagy such as mTOR, FOXO3, and microRNA *let-7* were significantly altered upon EtOH-induced stress (Fig 9), where mTOR inhibition may be accounted for by *PI3K* downregulation and reciprocal *PTEN* upregulation in EtOH-treated cells. Additionally, upregulation of tumor suppressors *TP53* and *CDKN2A* may suggest their roles in cell-cycle and DNA-damage check-point functions to prevent EtOH-induced aberrant growth and malignant transformation in esophageal carcinogenesis. Our RNA-seq data suggest that EtOH exposure may affect nuclear proteins essential in mitochondrial functions. These include TP53 and FOXO3 (Fig 9A and 9B). Under cellular stress, TP53 is translocated to mitochondria to interact with the Bcl2 family members to trigger apoptosis [49]. FOXO3 regulates transcription of oxidative phosphorylation (OXPHOS) genes in mitochondria. AMPK promotes FOXO3 translocation to mitochondria and mtDNA binding [50]. Thus, FOXO3 may contribute to maintenance of mitochondrial functions in cells surviving EtOH-induced stress conditions, warranting further study in future investigations.

AMPK is an essential intracellular energy sensor. Mitochondrial damage has been implicated in AMPK activation [8, 51]. In this study, AMPK activation was suggested in EPC2 cells by AMPK phosphorylation in the context of EtOH-induced mitochondrial dysfunction. Our findings reinforce the role for AMPK signaling in mitochondrial homeostasis and oxidative stress [12, 25, 52]. The tumor suppressor role of AMPK has been implicated in esophageal carcinogenesis and tumor progression [53–55]. Given AMPK activation in normal esophageal keratinocytes exposed to EtOH (Fig 10), it is tempting to speculate that AMPK may limit alcohol-induced esophageal carcinogenesis via autophagy by removing damaged and dysfunctional mitochondria that produce DNA-damaging superoxide. However, our findings from this study are limited to a context of acute alcohol exposure and a non-transformed, yet hTERT-immortalized cell line EPC2. Chronic alcohol exposure may decrease AMPK activity to promote hepatic steatosis that was rescued via pharmacological AMPK activation [52]. Further studies are required to understand the role of AMPK in esophageal cell injury and carcinogenesis that occur in the context of long-term and excessive alcohol consumption. Of note, one of the foremost target organs for alcohol-related carcinogenesis is the esophagus, where keratinocytes undergo malignant transformation to give rise to squamous cell carcinoma, which is the deadliest of all human squamous cell carcinomas and common worldwide [56]. Thus, extensive study is warranted to determine the functional consequences of long-term EtOH exposure in the esophageal epithelium, especially *in vivo*.

In conclusion, this study provides novel mechanistic insights into the role of alcohol in esophageal epithelial cell injury. In esophageal keratinocytes, EtOH exposure may lead to mitochondrial dysfunction, resulting in decreased ATP production and increased cellular ROS, and activation of AMPK. AMPK activation may possibly contribute to mTORC1 suppression to induce autophagy to maintain mitochondrial homeostasis and promote cell survival under EtOH-induced oxidative stress.

## Supporting information

**S1 Fig.**
(PDF)

**S2 Fig.**
(PDF)

**S3 Fig.**
(PDF)

**S4 Fig.**
(PDF)

**S5 Fig.**
(PDF)

**S6 Fig.**
(PDF)

**S7 Fig.**
(PDF)

**S8 Fig.**
(PDF)

**S9 Fig.**
(PDF)

**S1 Table.**
(PDF)

**S2 Table.**
(PDF)

## Acknowledgments

We thank Drs. Donita C. Brady, John P. Lynch, Jonathan P. Katz, Narayan G. Avadhani (University of Pennsylvania), and Anil K. Rustgi (Columbia University) for their critical input on the development of this project. We thank Dr. Balaraman Kalyanaraman (Medical College of Wisconsin) and Joy Joseph for their generous gift of Mito-CP. We thank Mr. Ben Rhoades and the staff of the Molecular Pathology and Imaging Core, Cell Culture/iPS Core and Flow Cytometry and Cell Sorting Facilities at the University of Pennsylvania, and the Shared Resources (Flow Cytometry, and Molecular Pathology, and Confocal & Specialized Microscopy) at the Herbert Irving Comprehensive Cancer Center at Columbia University for technical support. We thank members of the Nakagawa, Rustgi, Avadhani, and Baur labs for helpful discussions.

## Author Contributions

**Conceptualization:** Prasanna M. Chandramouleeswaran, Hiroshi Nakagawa.

**Data curation:** Prasanna M. Chandramouleeswaran, Manti Guha, Masataka Shimonosono, Kelly A. Whelan, Hisatsugu Maekawa, Uma M. Sachdeva, Gordon Ruthel, Sarmistha Mukherjee, Noah Engel, Michael V. Gonzalez, James Garifallou, Andres J. Klein-Szanto, Clementina A. Mesaros.

**Formal analysis:** Prasanna M. Chandramouleeswaran, Manti Guha, Masataka Shimonosono, Kelly A. Whelan, Uma M. Sachdeva, Gordon Ruthel, Sarmistha Mukherjee, Michael V. Gonzalez, James Garifallou, Andres J. Klein-Szanto, Clementina A. Mesaros.

**Funding acquisition:** Kelly A. Whelan, Ian A. Blair, Renata Pellegrino da Silva, Hakon Hakonarson, Eishi Noguchi, Hiroshi Nakagawa.

**Project administration:** Hiroshi Nakagawa.

**Resources:** Shinya Ohashi.

**Supervision:** Ian A. Blair, Renata Pellegrino da Silva, Hakon Hakonarson, Joseph A. Baur, Hiroshi Nakagawa.

**Writing – original draft:** Prasanna M. Chandramouleeswaran, Manti Guha, Masataka Shimonosono.

**Writing – review & editing:** Kelly A. Whelan, Uma M. Sachdeva, Andres J. Klein-Szanto, Eishi Noguchi, Hiroshi Nakagawa.

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
