## [Decision Letter · Decision Letter 0]

13 Jul 2020

PONE-D-20-13394

Autophagy mitigates ethanol-mediated mitochondrial damage and oxidative stress via the AMPK-mTORC1 axis in esophageal keratinocytes

PLOS ONE

Dear Dr. Nakagawa,

Thank you for submitting your manuscript to PLOS ONE. After careful consideration, we feel that it has merit but does not fully meet PLOS ONE’s publication criteria as it currently stands. Therefore, we invite you to submit a revised version of the manuscript that addresses the points raised during the review process.

ACADEMIC EDITOR: As per Reviewer 3, the issues regarding replicates must be addressed during the revision. 

We look forward to receiving your revised manuscript.

Kind regards,

Ravirajsinh Jadeja, Ph.D

Academic Editor

PLOS ONE

Journal Requirements:

2.  Please provide additional information about each of the cell lines used in this work, including the source of the cell lines and any quality control testing procedures conducted (authentication, characterisation, and mycoplasma testing). For more information, please see " ext-link-type="uri" xlink:type="simple">http://journals.plos.org/plosone/s/submission-guidelines#loc-cell-lines."

3. Please note that PLOS does not permit references to “data not shown.” Authors should provide the relevant data within the manuscript, the Supporting Information files, or in a public repository. If the data are not a core part of the research study being presented, we ask that authors remove any references to these data."

4. At this time, we ask that you please provide scale bars on the microscopy images presented in Figure 3 and refer to the scale bar in the corresponding Figure legend

Reviewers' comments:

Reviewer's Responses to Questions

**Comments to the Author**

1. Is the manuscript technically sound, and do the data support the conclusions?

Reviewer #1: Yes

Reviewer #2: Yes

Reviewer #3: Partly

2. Has the statistical analysis been performed appropriately and rigorously? 

Reviewer #1: Yes

Reviewer #2: Yes

Reviewer #3: No

3. Have the authors made all data underlying the findings in their manuscript fully available?

Reviewer #1: Yes

Reviewer #2: Yes

Reviewer #3: Yes

4. Is the manuscript presented in an intelligible fashion and written in standard English?

Reviewer #1: Yes

Reviewer #2: Yes

Reviewer #3: Yes

5. Review Comments to the Author

Reviewer #1: The esophagus is exposed to high concentrations of ethanol during alcohol consumption and this exposure is a risk factor for ESCC. Therefore, there is a critical need to understand the effects of alcohol on the esophageal epithelium. This manuscript investigates the induction of mitochondrial dysfunction subsequent to alcohol exposure in esophageal keratinocytes. The authors use cell culture-based systems to determine the lethal dose and duration of ethanol in their cells, identify that the mitochondria (and therefore the TCA) is dysregulated and that autophagy is a potential cellular mechanism these cells are using to mitigate the damage induced by the alcohol. They, furthermore, implicate AMPK and mTOR signaling in this mechanism of autophagy. The manuscript is fairly well written with a few grammatical errors noted below. The data support the conclusions of the paper, with the exception of mTOR pathway – again discussed below. The statistics are applied in an appropriate manner and aid inn a clear understanding of the data presented.

There are two significant points to address for this reviewer for this manuscript:

1) In describing the dilutions used for experimental conditions and controls, it is unclear if the media nutrients are more dilute in the alcohol-containing samples versus the control samples. In the methods it is mentioned that the alcohol is diluted in media while the control cells receive only media. It would be better if the experiments also had diluted control media to a similar extent as the alcohol-containing media. This could be clarified in the methods to more fully explain how the treatments occur if this interpretation is incorrect. Alternatively, this limitation could be explored in the discussion section.

2) It is unclear to this reviewer which experiments have been performed with multiple independent cell lines. The results section should clearly indicate when the data is collected in EPC1, EPC2, or EPC3 cell lines and the figures appropriately labelled for clarity.

Minor points to address:

1) Editing/grammar errors to fix in lines: 62, 69, 101, 283, 331, 492.

2) Should delete reference to in vivo data since none is presented.

3) How many reads were collected for the RNA-Seq data?

4) Line 248, how was apoptosis measured?

5) when quantifying data from organoids, how many were counted/slide and how many replicate slides were quantified?

6) The two bar graphs in figure 5A should receive their own lettering (ie 5b and 5c) and the other lettering adjusted appropriately.

7) the conclusion that mTORC1 signaling is altered from analyzing S6 phosphorylation is a bit of an overstated conclusion as stated in the subheading on line 397. The RNA-Seq analysis suggests the pathway as suppressed but one experimental piece of data was presented to support this. S6 can be phosphorylated by other pathways than mTORC1 signaling. Would suggest softening the conclusion in this section.

Reviewer #2: Authors propose that autophagy mitigates ethanol-mediated mitochondrial damage and oxidative stress via the AMPK-mTORC1 axis in esophageal keratinocytes.

This is a potentially interesting paper, however, several problems must be corrected.

1) Some statements are simply not correct such as

"RNA sequencing disclosed robust mitochondrial dysfunction".

RNA sequencing may suggest (!) mitochondrial dysfunction but never shows directly mitochondrial dysfunction.

You can change this to

"RNA sequencing shows mitochondrial alterations."

Please emphasize both in abstract and text that you have provided provide direct evidence for the mitochondrial dysfunction by following methods: Mitochondrial metabolism; Respiratory, Mitochondrial oxidative stress (MitoSOX) and cellular ATP level.

2) Please explain how ethanol induces mitochondrial stress (Figure 7, please revise to dysfunction). It is most likely not a direct effect. Please explain in introduction and discussion sections. RNA sequencing data suggest that ethanol may initially affect the nuclei protein expression which later translate into mitochondrial dysfunction. Please comment and explain.

Reviewer #3: This manuscript “Autophagy mitigates ethanol-mediated mitochondrial damage and oxidative stress via the AMPK-mTORC1 axis in esophageal keratinocytes” primarily aimed to investigate the effect of acute ethanol exposure on esophageal epithelial cells and its associated downstream signaling cascade. Authors had characterized the esophageal epithelial cells growth and viability at different time points in response to various concentrations of ethanol. Using RNA sequencing, LC-MS and TEM imaging, authors had showed mitochondrial dysfunction and structural changes in esophageal epithelial cells upon acute ethanol exposure. Authors had also found activation of the AMPK-autophagy axis in esophageal epithelial cells as a compensatory cytoprotective mechanism against alcohol induced cell injury.

Major comments:

1. The major concern of the manuscript is an inconsistency of cell lines used in each experiment and thus the statistical analysis. Most of the statistical analysis of the data was drawn from either one or 2 cell lines (except for RNA seq). It would be nice to show and draw the conclusions using 3 cell lines (n=3) in triplicates (as biological replicates).

2. In Fig.3B, authors had claimed that increase in intracellular vacuolar structures as reminiscent of autophagy vesicles (AV). However, the vacuoles in the TEM images look like having single membrane, whereas AV are double membrane structures.

3. Also, in Fig. 3B, authors had showed * Mitochondria lacking cristae (no changes in mitochondrial volume). Is acute ethanol exposure lead to complete collapse of cristae without any effect on mitochondria volume? Authors should comment on this.

4. Authors had claimed that involvement of mitophagy to regulate mitochondrial homeostasis to promote cell survival upon ethanol exposure. Authors had drawn this conclusion based on conversion of LC3BI to II form and p62 degradation (using single cell line, Fig 5A). It would be appropriate to show co-localization of PINK Parkin or mitochondrial marker with lysosomal marker as an indicator of mitophagy and its flux.

5. There is an inconsistence data regarding effect of 2% ethanol on cell viability within 8hr. Fig 1. explains no effect on cell viability, whereas Fig 5D showed effect on cell viability. Authors should comment on this variability.

6. No statistical analysis for some of the western blot data (Fig 6C, Fig 7A).

7. This study is an in vitro study. So, authors should modified the sentence in the introduction (line 8081).

6. PLOS authors have the option to publish the peer review history of their article (what does this mean?). If published, this will include your full peer review and any attached files.

Reviewer #1: No

Reviewer #2: No

Reviewer #3: No

---

## [Author Response · Author response to Decision Letter 0]

1 Sep 2020

Dr. Jadeja (Academic Editor)’s comment: “As per Reviewer 3, the issues regarding replicates must be addressed during the revision”.

We agree that conclusions would be strengthened with three cell lines as biological replicates and data expressed as an average of three. We have utilized EPC1, EPC2 and EPC3 to evaluate ethanol (EtOH) exposure in 3D organoids and monolayer culture (Figs 1 and 2). We have analyzed all three cell lines by RNA-seq to identify genes that are commonly altered upon EtOH exposure (Fig 3). To validate RNA-seq data, however, we focused upon EPC1 and EPC2 as biological replicates in the remainder of the manuscript. We chose EPC1 and EPC2 because they tolerated EtOH better than EPC3, as discussed (page 16, line 369-372). Gene expression patterns in EPC1 and EPC2 were deemed closer than EPC3 by principal component analysis (S4 Fig). EPC1 and EPC2 displayed consistent results regarding mitochondria morphology (Fig 4), mitochondrial membrane potential (new data, Fig 5), and autophagy (Fig 7 and Fig 8). We included additional data with EPC2 to complement the above findings. We have determined cellular ATP level and metabolic profiling (Fig 6), EtOH oxidation by ADH1B and CYP2E1 (new data, S5 Fig), and pharmacological modulations of AMPK (Fig 10B and C).

We observed differences between EPC1 and EPC2 cells about the effect of EtOH upon AMPK-mTORC1 axis, however. In EPC1, EtOH did not induce as robust changes in mTORC1 substrates (Fig 9C) and AMPK (Fig 10A) as compared to EPC2 cells. We clarified in the revised text that our model (Fig 11) was supported by data with EPC2 cells only. Additionally, further in vivo data, beyond demonstration of mitochondria depolarization in mice receiving EtOH (Fig 5D), are currently not available. These limitations require future investigation to reinforce the model, utilizing primary cell culture, genetically engineered mice, and human subjects. We stressed these points in revised page 26, line 629-647, and modified our conclusions (page 31, lines 762-766) as well as the title of this manuscript to reflect cell-line specific roles of the AMPK-mTORC1 axis.

In general, we had 3-4 technical replicates in all assays, and performed statistical analysis as described in the methods section and figure legends. However, we could not perform statistical analysis on densitometry of most western blots because we could not have multiple technical replicates (i.e. multiple blots from the same set of cell lysates) due to low protein yield from EPCs. We utilized EPC1 and EPC2 as two biological replicates and performed at least two independent experiments for each cell line with cell lysates prepared on different days (i.e. different sets of cell culture plates with or without treatments). Although we performed densitometry on all western blots, we did not show the data as an average of two from two independent experiments. We instead presented representative blots (Figs 7A, 9C and 10A) for each cell line with the densitometry data (S6 Fig).

Reviewer 1: “…The manuscript is fairly well written with a few grammatical errors noted below. The data support the conclusions of the paper, with the exception of mTOR pathway…discussed below.”

Major points:

1) “In describing the dilutions used for experimental conditions and controls, it is unclear if the media nutrients are more dilute in the alcohol-containing samples versus the control samples. In the methods it is mentioned that the alcohol is diluted in media while the control cells receive only media. It would be better if the experiments also had diluted control media to a similar extent as the alcohol-containing media. This could be clarified in the methods to more fully explain how the treatments occur if this interpretation is incorrect. Alternatively, this limitation could be explored in the discussion section”

In most experiments, cells were exposed to 2% ethanol (EtOH) for up to 8 h. To assess the impact of medium dilution, we compared media supplemented with or without phosphate buffered saline (PBS) in place of EtOH. PBS did not affect cell viability of all three esophageal cell lines tested (revised Figs 1B and 2A). We concluded that the potential effect of medium dilution is negligible, if any, as noted in the revised page 13, lines 302-304 and page 14, line 325.

In another experiment, now shown in S3 Fig, cell culture medium was first removed and cells were then exposed for 15 seconds to 5-80% EtOH in PBS or PBS only (control). Following removal of PBS with or without EtOH, cells were refed with regular cell culture medium and allowed to grow until cell viability was assessed at 8 h and 24 h time points. In this experiment, medium was not diluted by EtOH or PBS. We clarified the experimental condition in the revised legend for S3 Fig.

2) “It is unclear to this reviewer which experiments have been performed with multiple independent cell lines. The results section should clearly indicate when the data is collected in EPC1, EPC2, or EPC3 cell lines and the figures appropriately labelled for clarity.”

Please see our response to Academic Editor (Dr. Jadeja)’s comment above. In the revised results section, we have now clarified in text and figures which cell lines were used for each experiment.

Minor points:

1) Editing/grammar errors to fix in lines: 62, 69, 101, 283, 331, 492.

We have corrected these errors throughout the manuscript.

2) “Should delete reference to in vivo data since none is presented”.

We have now included in vivo data. We demonstrated that a subset of esophageal epithelial cells in mice display mitochondria depolarization following a single bolus oral gavage of EtOH (new Fig 5D) as described in revised page 5, lines 94-101 (methods) and page 18, lines 426-428 (results).

3) “How many reads were collected for the RNA-Seq data?”

We have obtained ~32 million input reads per sample on average of which ~24 million were uniquely mapped as described (revised page 15, lines 359-360). We have now included these data in S1 Table.

4) “Line 248 (revised page 14, lines 333-334), how was apoptosis measured?”

We have performed flow cytometry for Annexin V and propidium iodide-stained cells to determine apoptosis as shown in new S2 Fig.

5) “when quantifying data from organoids, how many were counted/slide and how many replicate slides were quantified?”

As elaborated in the revised legend for Fig 1 (page 13, lines 311-313), we generated organoids in triplicate per condition or time point, and performed live-cell quantitation by automated bright-field microscopy that imaged at least 100 organoids per well and determined the size of individual structures at each time point shown.

6) “The two bar graphs in figure 5A (revised Fig 7A) should receive their own lettering (ie 5b and 5c)…”

We agree and labeled these bar graphs accordingly after moving them to S6 Fig.

7) “the conclusion that mTORC1 signaling is altered from analyzing S6 phosphorylation is a bit of an overstated conclusion as stated in the subheading on line 397 (revised line 553). The RNA-Seq analysis suggests the pathway as suppressed but one experimental piece of data was presented to support this. S6 can be phosphorylated by other pathways than mTORC1 signaling. Would suggest softening the conclusion in this section”.

We agree and modified the conclusion (revised page 23, line 557) not to overstate about mTORC1 signaling. To reinforce the possible involvement of mTORC1 signaling, we have probed 4E-BP1 as an additional mTORC1 substrate in western blots (revised Fig 9C and S6 Fig). Following EtOH exposure, we observed robust inhibition of phosphorylation of the mTORC1 target S6 in both EPC1 and EPC2 cells, as well as decreased phosphorylation of the mTORC1 target 4EBP-1 in EPC2, but little if any in EPC1 cells. This likely reflects some differences in the mTORC1 downstream pathways between the two cell lines. We extended our discussions (revised page 26, lines 629-642).

Reviewer 2: “This is a potentially interesting paper, however, several problems must be corrected”.

Major points:

1) Some statements are simply not correct such as "RNA sequencing disclosed robust mitochondrial dysfunction"….RNA sequencing may suggest (!) mitochondrial dysfunction but never shows directly mitochondrial dysfunction…You can change this to "RNA sequencing shows mitochondrial alterations."

We agree that our original manuscript contained incorrect or bold statements. We appreciate the helpful advice by this Reviewer and have revised accordingly.

“Please emphasize both in abstract and text that you have provided direct evidence for the mitochondrial dysfunction by following methods: Mitochondrial metabolism; Respiratory, Mitochondrial oxidative stress (MitoSOX) and cellular ATP level”.

We have edited the abstract and the text to clarify the methods used to show mitochondrial dysfunction.

2) “Please explain how ethanol induces mitochondrial stress (Figure 7 [revised Fig 11], please revise to dysfunction). It is most likely not a direct effect. Please explain in introduction and discussion sections. RNA sequencing data suggest that ethanol may initially affect the nuclei protein expression which later translate into mitochondrial dysfunction. Please comment and explain”.

We have changed mitochondrial “stress” to “dysfunction” in revised Fig 11 as recommended by this Reviewer.

We agree that it is unlikely that ethanol (EtOH) induces mitochondrial dysfunction directly. It is well established that EtOH treatment results in conditions that promote mitochondrial ROS formation by mitochondria as corroborated by our data (Fig 8A and B). ROS have been shown to form mitochondrial (mt) DNA adducts and mtDNA damage upon EtOH exposure (Ref 44, PMID: 12055609). Additionally, EtOH affects the mitochondrial antioxidant mechanism, further exacerbating the damage (Ref 45, PMID: 11371722) as discussed in revised page 28, lines 701- 704.

We have explored the potential role of EtOH oxidation in mitochondrial dysfunction. To this end, we first assessed mitochondrial membrane potential (∆Ψm) by flow cytometry in EtOH-treated EPC1 and EPC2 cells (new Fig 5) where EtOH induced a small subset of cells with decreased ∆Ψm, indicating mitochondrial depolarization. We performed RNA interference experiments directed against ADH2 and CYP2E1, two major alcohol metabolizing enzymes that catalyze EtOH oxidation (new S5 Fig). We have observed that knockdown of either molecule prevented EtOH from inducing cells with mitochondrial depolarization, suggesting that EtOH oxidation may be involved in mitochondrial dysfunction. We have described these results (revised page 17, line 411-page 18, line 433; page 20, lines 489-493), and discussed (revised page 27, line 678-page 28, line 699).

It is most likely acetaldehyde, a toxic chief metabolite of EtOH, contributes to mitochondrial dysfunction. We have previously demonstrated that acetaldehyde causes oxidative stress in murine esophageal keratinocytes and that was augmented in cells with loss of Aldh2, a mitochondrial enzyme essential in acetaldehyde clearance (Ref 5, PMID: 27186430). Of three cell lines tested, EPC3 appeared to be more sensitive to the toxic effect of 2% EtOH than EPC1 and EPC2 (revised Fig 1B and C, and Fig 2A). EPC1 and EPC2 cells carry wild-type ALDH2 while EPC3 carries a single nucleotide polymorphism for ALDH2. This mutant ALDH2 enhances EtOH toxicity by delaying acetaldehyde clearance. We discussed the possibility that the ALDH2 status may influence the degree of EtOH-induced mitochondrial dysfunction (revised page 27, lines 657-671).

Our RNA-seq data suggest that EtOH exposure may affect nuclear proteins essential in mitochondrial functions. They include TP53 and FOXO3 (Fig 9A and B). Under cellular stress, TP53 is translocated to mitochondria to interact with the Bcl2 family members to trigger apoptosis (Ref 49, PMID: 19007744). FOXO3 regulates transcription of oxidative phosphorylation (OXPHOS) genes in mitochondria. AMPK promotes FOXO3 translocation to mitochondria and mtDNA binding (Ref 50, PMID: 29445193). Thus, FOXO3 may contribute to maintenance of mitochondrial functions in cells surviving EtOH-induced stress conditions. We have discussed these potential mechanisms in the revised manuscript (revised page 30, lines 733-739).

Reviewer 3:

Major comments:

1. “The major concern of the manuscript is an inconsistency of cell lines used in each experiment and thus the statistical analysis. Most of the statistical analysis of the data was drawn from either one or 2 cell lines (except for RNA seq). It would be nice to show and draw the conclusions using 3 cell lines (n=3) in triplicates (as biological replicates).”

Please see our response to Academic Editor (Dr. Jadeja)’s comment at the beginning of this rebuttal where we have addressed this important point raised by Reviewer 3.

2. “In Fig.3B, authors had claimed that increase in intracellular vacuolar structures as reminiscent of autophagy vesicles (AV). However, the vacuoles in the TEM images look like having single membrane, whereas AV are double membrane structures.”

We have rereviewed TEM images with an experienced pathologist (AJK) who has identified double-membrane vesicles as shown in revised Fig 4C. Most of the vacuoles were suspected as involutional mitochondria and some autophagy vesicles. The apparent lack of double membrane is possibly accounted for by insufficient resolution to detect distended (double) membranes in vacuolized (distended) organelles as discussed in revised page 29, lines 714-724).

3. “Also, in Fig. 3B, authors had showed * Mitochondria lacking cristae (no changes in mitochondrial volume). Is acute ethanol exposure lead to complete collapse of cristae without any effect on mitochondria volume? Authors should comment on this”.

In the original text (original page 13, line 307; revised page 16, line 393), we have pointed out volume increase (swelling) of mitochondria accompanying cristae changes. We have now clarified this in the revised legend for Fig 4B.

4. “Authors had claimed that involvement of mitophagy to regulate mitochondrial homeostasis to promote cell survival upon ethanol exposure. Authors had drawn this conclusion based on conversion of LC3BI to II form and p62 degradation (using single cell line, Fig 5A). It would be appropriate to show co-localization of PINK Parkin or mitochondrial marker with lysosomal marker as an indicator of mitophagy and its flux”.

We agree visualization of mitophagy is important as we have previously demonstrated in transformed EPC2 derivative cells under oxidative stress stimulated by transforming growth factor-β (Ref 18, PMID: 28414310). However, it was not our intention to claim involvement of mitophagy based on our western blot data in this study. We stated in the original Results section regarding transmission electron microscopy findings (Fig 4B) that “…numerous abnormal mitochondria were positioned adjoining to these vacuolar structures, suggesting that mitochondria-targeted autophagy (i.e. mitophagy) may be occurring…” This was a bold statement without direct evidence of mitophagy provided. We have removed this statement. We now reinforced our autophagy data in both EPC1 and EPC2 cells (revised Figs 7 and 8). In the discussion section, we have pointed out the possibility that autophagy may have a role in removal of EtOH-induced dysfunctional mitochondria via mitophagy, to promote cell survival under ethanol-induced stress, a subject that warrants further study in future investigations (revised page 29, lines 721-724).

5. “There is an inconsistence data regarding effect of 2% ethanol on cell viability within 8hr. Fig 1. explains no effect on cell viability, whereas Fig 5D showed effect on cell viability. Authors should comment on this variability”.

Original Fig 1 and Fig 5D showed differential experimental conditions in 3D organoids and monolayer culture, respectively. Additionally, we cited 3D organoid data (original Fig. 1C) as a rationale to use 2% ethanol (EtOH) and 8 h exposure time for RNA-seq experiments performed in monolayer culture. We apologize for any confusion resulting from our initial presentation of these data. We have now presented 3D organoid and monolayer culture data separately (revised Figs 1 and 2). Esophageal epithelial cells appeared to be more sensitive to EtOH-induced toxicity in monolayer culture than in tissue-like 3D organoid culture conditions (revised Figs 1B and 2A). We have cited monolayer culture data (revised page 14, lines 325-342; page 15, lines 357-358) as a basis of setting conditions for RNA-seq experiments.

We have also recognized variability, albeit modest, in EPC1 and EPC2 cell response to 2% EtOH. In revised Fig 8D and E, 2% EtOH alone did not affect EPC1 cell viability at 8 h time point. However, EPC2 cell viability was decreased by 10% as evaluated by WST-1 assays that detect live cells. In revised Fig 2B, 2% EtOH did not increase cell death in either EPC1 or EPC2 at 8 h time point as assessed by flow cytometry of DAPI-stained cells that detects dead, but not necessarily dying, cells. This likely reflects differences in the assay mechanisms and sensitivities, as noted in revised page 22, line 538. 

6. “No statistical analysis for some of the western blot data (Fig 6C, Fig 7A)”.

Please see our discussion at the end of our response to Academic Editor (Dr. Jadeja)’s comment above about this critical point and our revised efforts to address it.

7. “This study is an in vitro study. So, authors should modified the sentence in the introduction (line 8081)”.

We have now included in vivo data. We demonstrated that a subset of esophageal epithelial cells in mice display mitochondria depolarization following a single bolus oral gavage of EtOH (new Fig 5D) as described in revised page 5, lines 94-101 (methods) and page 18, lines 426-428 (results).

---

## [Editor Report · Decision Letter 1]

10 Sep 2020

Autophagy mitigates ethanol-induced mitochondrial dysfunction and oxidative stress in esophageal keratinocytes

PONE-D-20-13394R1

Dear Dr. Nakagawa,

We’re pleased to inform you that your manuscript has been judged scientifically suitable for publication and will be formally accepted for publication once it meets all outstanding technical requirements.

Kind regards,

Ravirajsinh Jadeja, Ph.D

Academic Editor

PLOS ONE

Additional Editor Comments (optional):

The Authors have done a substantial revision and the revised manuscript can be accepted in its present format.
---

## [Editor Report · Acceptance letter]

14 Sep 2020

PONE-D-20-13394R1

Autophagy mitigates ethanol-induced mitochondrial dysfunction and oxidative stress in esophageal keratinocytes

Dear Dr. Nakagawa:

I'm pleased to inform you that your manuscript has been deemed suitable for publication in PLOS ONE. Congratulations! Your manuscript is now with our production department.

Kind regards,

on behalf of

Dr. Ravirajsinh Jadeja 

Academic Editor

PLOS ONE